©c Author(s) 2019. CC BY 4.0 License.





# FORHYCS v1.0: A spatially distributed model combining hydrology and forest dynamics

Matthias J.R. Speich[1,2,3,4], Massimiliano Zappa[2], Marc Scherstjanoi[1,5], and Heike Lischke[1]

[1]Dynamic Macroecology, Swiss Federal Research Institute WSL, 8903 Birmensdorf, Switzerland
[2]Hydrological Forecasts, Swiss Federal Research Institute WSL, 8903 Birmensdorf, Switzerland
[3]Department of Environmental Systems Science, ETH Zurich, 8092 Zurich, Switzerland
[4]Biometry and Environmental Systems Analysis, University of Freiburg, 79085 Freiburg i. Br., Germany
[5]Institute of Climate-Smart Agriculture, Johann Heinrich von Thünen Institute, 38116 Braunschweig, Germany

**Correspondence:** Matthias Speich (matthias.speich@wsl.ch)

**Abstract.** We present FORHYCS (FORests and HYdrology under Climate Change in Switzerland), a distributed ecohydrological model to assess the impact of climate change on water resources and forest dynamics. FORHYCS is based on the coupling of the hydrological model PREVAH and the forest landscape model TreeMig. In a coupled simulation, both original models are executed simultaneously and exchange information through shared variables. The simulated canopy structure is summarized

by the leaf area index (LAI), which affects local water balance calculations. On the other hand, an annual drought index is obtained from daily simulated potential and actual transpiration. This drought index affects tree growth and mortality, as well as a species-specific tree height limitation. The effective rooting depth is simulated as a function of climate, soil and simulated above-ground vegetation structure. Other interface variables include stomatal resistance and leaf phenology.

Case study simulations with the model were performed in the Navizence catchment in the Central Swiss Alps, with a sharp

elevational gradient and climatic conditions ranging from dry inneralpine to high alpine. In a first experiment, the model was run for 500 years with different configurations. The results were compared against observations of vegetation properties from national forest inventories, remotely sensed LAI and high-resolution canopy height maps from stereo aerial images. Two new metrics are proposed for a quantitative comparison of observed and simulated canopy structure. In a second experiment, the model was run for 130 years under idealized climate change scenarios: daily temperature was increased by up to 6 K, and

precipitation altered by 10 %, with a gradual change over 35 years.

The first experiment showed that model configuration greatly influences simulated vegetation structure. In particular, simulations where height limitation was dependent on environmental stress showed a much better fit to canopy height observations. Spatial patterns of simulated LAI were more realistic than for uncoupled simulations of the forest landscape model, although some model deficiencies are still evident. Under idealized climate change scenarios, the effect of the coupling varied regionally,

with the greatest effects on simulated streamflow (up to 40 mm y$^{-1}$ difference with respect to a simulation with static vegetation parameters) seen at the valley bottom and in regions currently above the treeline. This case study shows the importance of coupling hydrology and vegetation dynamics to simulate the impact of climate change on ecosystems. Nevertheless, it also highlights some challenges of ecohydrological modelling, such as the need to realistically simulate plant response to increased $CO_2$ concentrations, and process uncertainty regarding future land cover changes.





# 1  Introduction

Of the manifold effects of climate change, many are expected to impact the interactions between the water cycle and forest dynamics. As a result of higher temperatures and shifts in precipitation regimes, an increase in the frequency and intensity of drought events is predicted (Allen et al., 2010), as experienced in Europe in 2003 and 2018. This may greatly affect tree growth

and mortality, even in locations currently not subject to high water stress (Choat et al., 2012; Martin-Benito and Pederson, 2015). This affects hydrologically relevant vegetation properties such as leaf area index (LAI) (Tesemma et al., 2015), root depth or biomass (Bréda et al., 2006) and stomatal conductance, which vary with stand age or species composition (Ewers et al., 2005; Ford et al., 2011). These changes might affect streamflow, but also feed back on local conditions for growth by altering water availability. For example, various studies have shown that trees growing in thinner stands are subject to lower

water stress, so that artificial thinning may mitigate drought effects on tree growth and mortality (Elkin et al., 2015) and increase water yield (McLaughlin et al., 2013). Also, hydrological sensitivity of catchments to climate change seems to depend on vegetation properties, with mixed forests showing a more stable water yield than catchments dominated by broadleaf or coniferous forests (Creed et al., 2014). Furthermore, increased atmospheric $CO_2$ is expected to impact hydrology through its effects on stomatal activity and plant productivity (Trancoso et al., 2017), although the long-term effects are still subject to

high uncertainty and debate (Medlyn et al., 2011). Another ecohydrologically relevant component of global change is land-cover change, driven by change of human land use, but also by natural vegetation dynamics. Increases in forested area usually reduce streamflow (Andréassian, 2004; Bosch and Hewlett, 1982), although the magnitude of such changes varies strongly with catchment characteristics such as climate, soil and forest age (Andréassian, 2004). Also, catchment response is often non-stationary, especially in the case of afforestation or reforestation, where streamflow strongly depends on stand age (Farley

et al., 2005). Recent model developments have aimed at improving predictions by including transient vegetation parameters to simulate the transition between forest and non-forest (Du et al., 2016; Nijzink et al., 2016).

Mountainous regions are particularly sensitive to global change (Mayor et al., 2017). Given the high significance of mountains for large-scale water supply (Viviroli et al., 2003), it is crucial to estimate how the factors affecting water supply will change in the future. In Switzerland, mountains make up two thirds of the territory and are of high importance for nature con-

servation, energy production, tourism and farming, among other sectors (SCNAT, 2012). Modeling studies predict a change in runoff regime due to increased temperatures, changing precipitation seasonality and glacier melt (Rössler et al., 2014), with important, but locally varying consequences for hydropower generation (Gaudard et al., 2014) and farming (Milano et al., 2015). Climate impacts on forests also vary locally, with an increase of drought stress predicted at lower elevations and improved growth conditions in energy-limited high-altitude forests (Bugmann et al., 2014), leading to shifting spatial patterns of species

composition, tree biomass and canopy cover (Bugmann et al., 2014; Fuhrer et al., 2006). The increased frequency of extreme droughts will probably be a more important factor than a change in long-term averages (Fuhrer et al., 2006). Additionally, abandonment of high-mountain pastures, driven by socio-economic processes, is an important factor of land-use change (Price et al., 2016), interacting with climate change to allow the tree line shift upwards (Gehrig-Fasel et al., 2007). These develop-





ments and predictions highlight the need for an integrated simulation of hydrology, forest dynamics and land use change in Switzerland.

## 1.1 Coupled models of hydrology and forest dynamics

Interactions between hydrology and vegetation dynamics are included in various types of dynamic models, with widely different areas of application, levels of complexity, and spatial and temporal resolutions (Fatichi et al., 2016). One such domain are land surface models (LSMs), which represent land surface processes in climate models (e.g. CLM; Lawrence et al., 2011). The main role of vegetation in these models is the partitioning of energy between sensible and latent heat fluxes. The latter consist of evaporation and transpiration, thus representing the coupling between vegetation, hydrology and the atmosphere. Over time, the representation of vegetation and evaporative fluxes in LSMs grew increasingly complex, moving from a simple vegetation-independent bucket model in early applications to a detailed description of vegetation processes, in particular physiological processes such as photosynthesis, carbon assimilation and nutrient cycling (see review by Seneviratne et al., 2010). For a representation of vegetation dynamics, climate models are sometimes coupled with dynamic global vegetation models (e.g. LPJ; Sitch et al., 2003), which can also be used for offline simulations. Although the primary objective of these models is to simulate vegetation patterns, they usually include a representation of terrestrial water balance, and are able to simulate river discharge (Gerten et al., 2004).

While these models typically operate at the global scale, and for computational efficiency simplify vegetation to a single average individual per plant type and coarse grid cell, some ecosystem models have been developed to depict vegetation structure, such as the dynamic vegetation model LPJ-GUESS (Smith et al., 2001), which is based on individual plants. Like LPJ, this model contains a soil hydrology module that can be used to calculate discharge, but such predictions were greatly improved by adding a coupling to a routing scheme (Tang et al., 2013). A sensitivity analysis by Pappas et al. (2013) pointed out that, despite of its detailed mechanistic representation of transpirational demand and stomatal closure that affects carbon uptake, water stress effects are represented inaccurately in LPJ-GUESS. As the various effects of water shortage on trees are well documented (McDowell et al., 2008), this may be seen as a weakness of LPJ-GUESS, especially since important plant functions, like cavitation or leaf area reduction under drought conditions are not implemented yet (Pappas et al., 2013; Manusch et al., 2014).

Another type of models representing the interface of hydrology and vegetation are forest water balance models. These usually simulate local water balance of forest stands to predict the influence of climatic change or forest management practices on growth conditions for trees. Examples include WAWAHAMO (Zierl, 2001) or BILJOU (Granier et al., 1999). These models may be coupled with dynamic forest models (Lischke and Zierl, 2002; Seely et al., 2015), or run with vegetation parameters assimilated from forest inventories (Zierl, 2001; De Cáceres et al., 2015) or remote sensing (Chakroun et al., 2014). Moreover, most forest dynamics models include a water balance module (Bugmann and Cramer, 1998; Seidl et al., 2012). However, its role is usually restricted to quantifying soil moisture stress, and it is seldom used to predict streamflow.

Various hydrological models operating at catchment scale have been used to evaluate the effect of land use change, and the resulting change in vegetation, on streamflow. Such models include distributed models with a high spatial resolution, allowing



for a detailed mapping of static vegetation parameters, e.g. DHSVM (Wigmosta et al., 1994). Other models include a simple vegetation growth module, such as SWAT (Watson et al., 2008) or SWIM (Wattenbach et al., 2005). More complex models include a detailed representation of average plant biomass growth, carbon and nutrient cycling, and hydrological processes, such as RHESSys (Tague and Band, 2004), or Tethys-Chloris (Fatichi et al., 2012).

Furthermore, the impact of vegetation dynamics on streamflow has also been studied with models that were not originally developed to this effect. For example, Sutmöller et al. (2011) used the physically-based, distributed hydrological model WaSiM-ETH (Schulla, 2015) to perform hydrological simulations with vegetation parameters derived from an individual based forest model driven by different forest management scenarios, which influence forest structure and species composition. Also, Schattan et al. (2013) applied the semi-conceptual hydrological model PREVAH (Gurtz et al., 1999) with vegetation parameters

obtained from the forest landscape model TreeMig (Lischke et al., 2006). In the original versions of both these hydrological models, vegetation parameters were parameterized as a function of season and land cover class only. This one-way coupling impacted mean annual streamflow by about 10 mm year$^{-1}$ in two large catchments in Switzerland, whereas the effect on local water balance reached 40 mm year$^{-1}$ in individual cells. Similarly, a modeling experiment by Köplin et al. (2013) showed that including transient land cover changes, such as forest cover increase or decrease or glacier retreat, could substantially affect

water balance predictions. Such experiments highlight the importance of considering the dynamics and spatial variability of vegetation properties in hydrological simulations.

So far, most of the models combining hydrology and vegetation processes have a strong biogeochemical focus, whereas successional dynamics and interspecific competition are rarely considered. Couplings between models that explicitly simulate forest dynamics and hydrological models have so far mostly been the object of experimental studies. The results from these

experiments show that coupling these processes may substantially alter model results and behavior (Sutmöller et al., 2011; Schattan et al., 2013). This gives an opportunity to increase the confidence in simulated impacts of climate change on forested ecosystems. For reliable, country-level predictions of long-term climate impacts on water resources and forest structure and composition in a mountainous country such as Switzerland, a model should:

- explicitly simulate the feedbacks between forest properties and hydrology;

- return estimates of streamflow and (evapo)transpiration as well as tree species distribution and biomass;

- operate at a spatial resolution fine enough to account for the great variability in climate, topography and land cover;

- be able to simulate the hydrology of non-vegetated areas such as glaciers, bare rock or built-up

- take into account the possibility of forest expansion and retreat

- not be too complex, so that it can be run for large areas and long periods at a reasonable computational cost.

None of the models discussed above fulfil all these criteria. This motivated the development of a spatially distributed model combining hydrology and forest dynamics, presented below.



## 1.2 Aims of this work

In this paper, we present a newly developed distributed ecohydrological model, FORHYCS (FORests and HYdrology under Climate change in Switzerland). This model combines two existing models, the hydrological model PREVAH (Gurtz et al., 1999) and the forest landscape model TreeMig (Lischke et al., 2006). FORHYCS is spatially distributed and operates on a

grid of regular cells. The model outputs are hydrological quantities, such as catchment-integrated streamflow and maps of runoff, transpiration, evaporation or snow cover, and maps of forest properties, such as biomass, leaf area index (LAI), species distribution and tree density. In FORHYCS, the two source models are run simultaneously, and exchange information through shared variables. FORHYCS may be run in uncoupled mode (i.e. parallel simulations of hydrology and forest dynamics, without any transfer of information between the two source models), with a one-way coupling (transfer of information from

the forest landscape model to the hydrological model, or vice-versa), or in fully coupled mode.

Like its parent models, FORHYCS may be described as semi-conceptual. Hydrological processes (evaporation, transpiration, soil moisture dynamics and runoff generation) and forest dynamics (growth, mortality, establishment and migration of tree species) are based on physical and ecological theory as well as empirical approaches. Thus, the degree of complexity of FORHYCS is lower than other coupled ecohydrological models, such as RHESSys (Tague and Band, 2004). Also, unlike these

models, the focus of the ecological part is on forest dynamics, similarly to the addition to SWIM proposed by (Wattenbach et al., 2005). However, FORHYCS differs from the latter approach in that growth and mortality are simulated at the level of species and size classes, instead of a single biomass pool per cell. Both TreeMig and PREVAH were designed for applications at an intermediate spatial resolution, with a cell size between 100 m and 1 km.

The goal of this paper is to explore the interplay of hydrology and forest dynamics in a coupled model. The questions to be

answered are (1) How does the coupling impact the behavior and the performance of both TreeMig and PREVAH, compared to uncoupled simulations? (2) Which aspects of the forest-hydrology coupling are of greatest importance for simulation results? (3) What are the implications of model coupling for simulations under climate change?

The model is tested in five subcatchments of the Navizence catchment (27 to 87 km$^2$), located in the Swiss Central Alps. To answer the first question, a full forest succession is modeled for the period 1500-2015, starting from bare soil. The forest

model is run in uncoupled mode to serve as a reference simulation. For the coupled runs, various configurations are tested, with different aspects of the coupling between hydrology and forest dynamics switched on and off. The outputs are then compared to forest inventory data, as well as gridded datasets of leaf area index (LAI) and canopy height. To assess the effect of the coupling on hydrological predictions, simulated streamflow from coupled and uncoupled model runs is compared against a time series of daily measurements. Furthermore, to assess the behavior of the coupled and uncoupled model under climate

change, the model is run for a century under artificial climate change scenarios. In these future model runs, the significance of two further aspects of the forest-hydrology coupling are examined: the effect of elevated $CO_2$ on stomatal resistance, as well as the potential forest expansion into high-elevation meadows as a result of climate and land-use change.





## 2   Methods and Data

### 2.1   FORHYCS model description

The two original models operate at a different temporal resolution: while TreeMig simulates forest dynamics with an annual time step, PREVAH calculates daily values, with an internal time step of one hour. Both original models are run simultaneously over the same domain and exchange information through shared variables. Figure 1 a) shows the flow of a simulation: hydrology is simulated on a daily basis with vegetation properties given by the forest model for the previous year. This is based on the assumption that effects of current-year water balance on the forest structure and composition are negligible. An annual drought index, influencing tree growth and mortality (Sect. 2.1.3), is calculated in each cell from the transpiration simulated in the hydrological model over the whole simulation year (Sect. 2.1.4). Based on the number of trees per species and height class in each cell, annual maximal values of Leaf Area Index (LAI) and fractional canopy cover (FCC) are calculated. These values are converted to daily values using a temperature-dependent phenology module (Sect 2.1.2). The root-zone water storage capacity (SFC) is updated annually as a function of long-term climate (Sect. 2.1.5). Furthermore, the model includes the effects of snow-induced mortality for some species using an additional mortality function based on snow cover duration (Sect. 2.1.6).

### 2.1.1   Source models

The semi-conceptual hydrological model PREVAH (Gurtz et al., 1999) in its fully distributed form (Schattan et al., 2013; Speich et al., 2015) solves the water balance of each grid cell by calculating evapotranspiration, soil water balance and runoff generation at a sub-daily time step. The core of PREVAH is based on the structure of the widely used model HBV (Bergström, 1992), combined with a conceptual runoff routing scheme (Gurtz et al., 1999). Successive model developments enabled or improved the treatment of interception (Menzel, 1996), snowpack dynamics (Zappa et al., 2003), glacier runoff (Klok et al., 2001) and groundwater runoff (Gurtz et al., 2003). PREVAH was used, among other applications, to estimate the impact of climate change on discharge (Zappa and Kan, 2007). Climate impact studies were also conducted using the distributed outputs of PREVAH (Speich et al., 2015).

The spatially explicit forest landscape model TreeMig (Lischke et al., 2006) simulates forest establishment, growth and mortality, as well as seed dispersal. Originally based on a gap model (Lischke et al., 1998), TreeMig calculates the number of trees per species and height in each cell, and operates at an annual time step. Inter- and intra-specific competition is represented through light distribution within the canopy, which depends on the distribution of trees of different height within a stand, and thus on leaf area (Lischke et al., 1998). TreeMig was used to predict climate impact on tree species distribution in Switzerland (Bugmann et al., 2014), as well as to simulate forest response to land abandonment (Rickebusch et al., 2007) and the feedbacks between forests and avalanches (Zurbriggen et al., 2014). Abiotic drivers of forest dynamics are represented by three bioclimatic indices: mean temperature of the coldest month, degree-day sum and drought (see Sect. 2.1.3 for the latter).





**Figure 1.** (a) Flow of a FORHYCS simulation starting in the example year 1971. The initial state can be loaded from a file, or a spin-up can be performed. The hydrological calculations are performed daily. At the end of the simulation year, bioclimatic indices are calculated, and passed to TreeMig, which simulates forest dynamics with an annual time step. Long-term climate statistics are updated at the end of the year based on the daily calculations, and the effective rooting depth module is run with the updated climate indices. The forest model returns an annual maximal LAI value. To be used in hydrological calculations, this value is converted to daily values based on simulated leaf phenology. (b) Schematic overview of the couplings between climate, above-ground forest structure and the rooting zone in TreeMig (left) and FORHYCS (right). TreeMig's forest structure (species-size distribution) depends on its previous state, and is influenced by the annual bioclimatic indices. Unlike the other indices, which depend on meteorological input only, the drought index DI further depends on a constant soil moisture storage capacity ("bucket size"). In FORHYCS, the drought index further depends on canopy structure, through its influence on potential transpiration (PT). Additional optional couplings in FORHYCS are the dynamic simulation of a climate-dependent storage capacity of the rooting zone (through varying effective rooting depth), a limitation of maximum tree height under dry conditions, as well as a drought-dependent reduction of leaf area.



### 2.1.2 Canopy structure and leaf phenology

Leaf Area Index (LAI) is calculated in TreeMig to account for mutual shading and competition for light (Lischke et al., 1998). In FORHYCS, LAI is passed to the hydrological model, replacing the land cover-specific parameterization in PREVAH. The allometric equations of Bugmann (1994) relate leaf area to diameter at breast height ($D$ [cm]). As $D$ is allometrically linked to tree height, leaf area is calculated for each group of trees of the same species and height class, then summed to obtain the value for the whole stand:

$$A_l = \sum_{sp=1}^{nspc} \sum_{hc=1}^{nhcl} SLA_{sp} \times a_{1,sp} \times D_{sp,hc}^{a2,sp} \times p_{d,sp}, \tag{1}$$

where $SLA_{sp}$ is specific leaf area [m² kg⁻¹], $a_{1,sp}$ [kg cm⁻¹] and $a_{2,sp}$ [-] are species-specific allometric parameters (Bugmann, 1994), and $p_{d,sp}$ is a reduction term accounting for seasonal variations in leaf area (see below and in the supplement). The reference area in TreeMig is the size of a forest plot, assumed to be 833 m². Therefore, LAI is calculated as:

$$LAI = A_l k_c/833, \tag{2}$$

where $k_c$ is a factor to convert from true to projected leaf area, set to 0.5 for broadleaves and 0.4 for conifers following Hammel and Kennel (2001). Fractional canopy cover $f_c$ is also calculated from the number of trees per species and height class, using the equation of Zurbriggen et al. (2014):

$$f_c = \left[1 - exp(-1 \times \sum_{sp=1}^{nspc} \sum_{hc=1}^{nhcl} (n_{sp,hc}/833) \times CA_{sp,hc})\right], \tag{3}$$

where $CA_{sp,hc}$ is the total crown area for trees of each species and height class:

$$CA_{sp,hc} = (k_{a1,sp} \times h_{hc}^2 + k_{a2,sp} \times h_{hc}) \times n_{sp,hc}, \tag{4}$$

where $k_{a1,sp}$ and $k_{a2,sp}$ are species-specific allometric parameters, and $h_{hc}$ the (upper) height of height class $hc$. Minimum values for LAI and $f_c$ are set to 0.2 and 0.1, respectively, to account for a minimal cover by grass and herbs, as well as bare stems. Annual maximum LAI may be reduced as a function of drought stress of the previous years, as described below in Section 2.1.3.

To obtain daily values for LAI and $f_c$, a leaf phenology module has been implemented. The variable $p_{d,sp}$ reflects the phenological status of species $sp$ in a given cell, and varies between zero (no leaves) and one (full foliage). This module also defines the start and end of the growing season in each cell, which is required to calculate the climatic indices for the rooting depth module (Sect. 2.1.5). In each cell, the growing season lasts as long as $p_{d,sp}$ is greater than 0.5 for the dominant species.





Daily values of $p_{d,sp}$ are simulated in spring with the model of Murray et al. (1989), which depends on the number of chill days in winter and of accumulated growing degree days in summer. When $p_{d,sp}$ reaches one, the foliage is assumed to be fully developed. The onset of leaf senescence in autumn is simulated using the model of Delpierre et al. (2009), which depends on temperature and photoperiod. For broadleaves, the end of the growing season is set 14 days after the onset of senescence,

and $p_{d,sp}$ is linearly reduced from one to zero during this period. Although the leaf area is not varied throughout the year for evergreen conifers, $p_{d,sp}$ is still simulated to define the start and end of the growing season. For these species (as well as for the deciduous conifer *Larix decidua*), the development of $p_{d,sp}$ following the onset of senescence is calculated using the formulation of Scherstjanoi et al. (2014). The parameters for the empirical models of Murray et al. (1989) and Delpierre et al. (2009) were calibrated against phenological observations across Switzerland (Defila and Clot, 2001). The species-specific

parameters and a description of the calibration procedure are given in the supplement.

### 2.1.3 Drought and its effects on forest growth, mortality and structure

The drought index used in TreeMig is based on the ratio of annual to potential evapotranspiration, as calculated with a module modified from Thornthwaite and Mather (1957). This scheme requires monthly precipitation sum and temperature average for each cell, as well as the plant-available water storage capacity. Evaporative demand is calculated using an empirical,

temperature-dependent approach, and monthly soil water balance is simulated based on water supply and demand, after accounting for interception (see Bugmann and Cramer, 1998, for a full description of this scheme). This approach has a number of drawbacks. First, it does not consider the effect of variations in vegetation properties on evaporative supply and demand, thus neglecting feedbacks between vegetation density, transpiration and drought (see e.g. Kergoat, 1998). Also, the Thornthwaite-Mather routine does not account for snow-related processes, which can lead to large errors in the estimation of evapotranspi-

ration, diminishing the representativeness of the index for growth conditions (Anderegg et al., 2013). Furthermore, empirical evapotranspiration formulations such as the Thornthwaite-Mather formula rely to a large extent on calibrated values. These may not be transferrable to climatic conditions that differ from the calibration period (Bartholomeus et al., 2015), limiting the ability of the model to make predictions under a changing climate. For these reasons, the EDI was replaced by a relative transpiration index (Speich et al., 2018a). This index is similar to the evapotranspiration deficit index presented above, but is

based on transpiration rather than evapotranspiration:

$$DI = 1 - \frac{\sum_{d=de}^{ds} E_{T,act,d}}{\sum_{d=de}^{ds} E_{T,pot,d}}, \tag{5}$$

where $de$ and $ds$ are the first and last day of the period for which the drought stress is calculated. Here, $DI$ is calculated for the entire year, so that $de = 1$ and $ds = 365$. Furthermore, to account for delayed effects of drought on tree physiology (Hammel and Kennel, 2001), the average of the last three years is used here. Transpiration is reduced from potential rate through the

effect of low soil moisture or high atmospheric vapor pressure deficit (VPD) on stomatal resistance (Eq. 14 in Speich et al. (2018a)). The rationale behind this index is based on the fact that stomatal closure is one of the first responses of a plant to water deficit. Therefore, the time during which stomatal resistance is increased due to drought is the time during which adverse





physiological effects of water shortage (e.g. cavitation, reduced carbon uptake) are likely to occur, and DI serves as a proxy for all these processes. The effect of VPD was included to account for the effects of high evaporative demand on plant-internal hydraulics (Zierl, 2001). The drought stress function in TreeMig determines the relative drought-induced limitation of annual growth (Bugmann, 1994):

$$f_{DS} = \sqrt{max(0, 1 - \frac{DI}{k_{DT}})},$$ 
(6)

where $k_DT$ is a species-specific drought tolerance parameter, indicating the value of $DI$ at which growth is completely suppressed. This growth reduction function can take values between zero (complete growth suppression) and one (unstressed conditions). Annual growth of the trees of the same species and height class is the product of a species-specific maximal growth and an environmental reduction function $f_{env}$. This reduction function is the geometric mean of $f_{DS}$ and two other stress functions, representing the effects of temperature and shading (Bugmann, 1994). The same reduction function is used to simulate mortality in addition to background mortality, and applies if it is more severe than mortality caused by low productivity. Lischke and Zierl (2002) parameterized $k_{DT}$ for the 30 tree species represented in TreeMig by overlaying modelled $DI$ with inventory-derived maps of species distribution. The values range between 0.27 and 0.5. However, this parameterization did not lead to satisfactory simulations of species composition in the case study of this paper. Therefore, species-specific $k_{DT}$ was defined based on a combination of the rankings by Lischke and Zierl (2002) and Niinemets and Valladares (2006). Table A1 lists the $k_{DT}$ values used in this study.

FORHYCS accounts for two additional effects of drought stress: a limitation of maximum height and a reduction of annual maximal LAI. The former is parameterized following Rasche et al. (2012), i.e. with a decrease of LAI with $DI$ down to a species-specific parameter $k_{redmax}$, indicating the fraction of species-specific maximum height that can be attained by a species if $DI$ is at $k_{DT}$. The same is done for annual degree-day sum, and the more severe of the two reductions is applied. Unlike in the formulation of Rasche et al. (2012), where the reduction is a linear function of the bioclimatic indices, the impact functions (Eq. 6 for drought and S8 for degree-day sum) are used here.

The LAI reduction function follows the formulation of Landsberg and Waring (1997), where the fraction of carbon allocated to roots increases under stress, whereas allocation to foliage and stem decreases. Since allocation is not explicitly simulated in FORHYCS, the following formulation is purely phenomenological. For all size classes of a given species, leaf area is scaled by the ratio of the foliage allocation coefficient under current $\eta_l$ and unstressed conditions $\eta_{l,u}$. Eq. 1 is thus modified as follows:

$$A_l = \sum_{sp=1}^{nspc} \sum_{hc=1}^{nhcl} SLA_{sp} \times a_{1,sp} \times D_{sp,hc}^{a2,sp} \times p_{d,sp} \times \eta_{l,sp} / \eta_{l,u,sp},$$ 
(7)

The allocation coefficients for foliage are calculated as:

$$\eta_l = 1 - \eta_r - \eta_s, \text{ and } \eta_{l,u} = 1 - \eta_{r,u} - \eta_{s,u},$$ 
(8)





where $\eta_r$ and $\eta_{r,u}$ are the allocation coefficients to roots, and $\eta_s$ and $\eta_{s,u}$ he allocation coefficients to the stem, under current and unstressed conditions, respectively. Following Landsberg and Waring (1997), $\eta_{r,u}$ is set to 0.229, and $\eta_r$ increases with increasing stress by the following relation:

$$\eta_r = \frac{0.8}{1 + 2.5(1 - fenv)}, \tag{9}$$

where $f_{env}$ is the geometrical mean of the drought and low temperature stress functions (Eqs. 6 and S8). The carbon allocated to the stem is related to $\eta_r$ as follows:

$$\eta_s = (1 - \eta_r)/(p_{l,s} + 1), \text{ and } eta_{s,u} = (1 - \eta_{r,u})/(p_{l,s} + 1), \tag{10}$$

where $p_{l,s}$ is the ratio of the growth rates of leaves and stems, in terms of their change in relation to diameter at breast height $D$. In FORHYCS, $p_{l,s}$ is calculated using the allometric equations used to calculate leaf and stem biomass:

$$p_{l,s} = \frac{dw_l/dD}{dw_s/dD} = \frac{k_{l,1} D^{k_{l,2}}}{k_{s,1} D^{k_{s,2}}}, \tag{11}$$

where $k_{l,1}$ and $k_{l,2}$ are allometric parameters for leaf biomass, and $k_{s,1}$ and $k_{s,2}$ are allometric parameters for stem biomass (Bugmann, 1994).

### 2.1.4   Partitioning of transpiration and soil evaporation

The implementation of the new drought index (see Sect. 2.1.3 above) required some changes to the evapotranspiration routine
in the hydrological model. While the relative transpiration index is based on estimates of actual and potential transpiration, PREVAH does not explicitly differentiate between transpiration and soil evaporation. Therefore, a new local water balance routine was implemented, based on the stand-alone model FORHYTM (Speich et al., 2018a). This module combines the soil water balance formulation of the HBV model (Bergström, 1992), which is also implemented in PREVAH, with the transpiration and evaporation scheme of Guan and Wilson (2009) and a Jarvis-type (Jarvis, 1976) parameterization of canopy resistance. A
full description is given in (Speich et al., 2018a).

The parameterization of canopy resistance differs from the original formulation in two ways. First, the effect of atmospheric vapor pressure deficit (VPD) on stomatal conductance is represented with a negative exponential function instead of a linear function. Second, an additional canopy resistance modifier ($f_5$) was implemented, to account for the effect of atmospheric $CO_2$ concentration ($C_a$ [$\mu mol\, mol^{-1}$]). This function is based on the results of Medlyn et al. (2001):

$$f_5 = \left(1 - (1 - j_c)(\frac{min(C_a, 700)}{350} - 1)\right)^{-1}, \tag{12}$$

where $j_c$ represents the fractional change in conductance in response to an increase in $C_a$ from 350 to 700 $\mu mol\, mol^{-1}$, and was set to 0.1 for coniferous forests, 0.25 for broadleaf forests and 0.18 for mixed forests (the forest type in each cell





is determined based on the relative share of above-ground biomass belonging to conifers and broadleaves). This affects both potential and actual transpiration, so that with all other factors kept constant, increases in $C_a$ will reduce the level of drought stress. The rationale for implementing this new water balance scheme is to account for the effect of variations in vegetation properties (e.g. LAI) on physiological drought in forests. As FORHYCS includes the possibility of changing land cover classes in a cell, some non-forested cells may become forested over the course of a simulation. As vegetation parameters (such as LAI and effective rooting depth) are prescribed as a function of land cover for non-forested cells, this shift inevitably introduces an artificial discontinuity in the simulation. To reduce this discontinuity, the new water balance scheme is also used for potentially forested land cover types. On the other hand, for land cover types that cannot become forested, the original water balance scheme of PREVAH (Gurtz et al., 1999) is applied.

### 2.1.5 Rooting zone storage capacity

The rooting zone water holding capacity, SFC, is calculated as the product of effective root depth $Z_e$ and soil water holding capacity $\kappa$ (Federer et al., 2003). While $\kappa$ is assumed to remain constant, $Z_e$ is assumed to vary as a function of vegetation characteristics and climate. The approach used to parameterize $Z_e$ is the carbon cost-benefit approach of Guswa (2008, 2010). This approach rests on the assumption that plants dimension their rooting systems in a way that optimizes their carbon budget. The optimal rooting depth is the depth at which the marginal carbon costs of deeper roots (linked to root respiration and construction) starts to outweigh the marginal benefits (i.e. additional carbon uptake due to greater availability of water for transpiration). The implementation of this model in FORHYCS follows the procedure described by Speich et al. (2018b). Effective rooting depth, expressed as an average over the whole cell, is calculated for both overstory (trees) and understory (shrubs and non-woody plants). The storage volume SFC for a given cell is defined as the sum of these two area-averaged rooting depths, multiplied with soil water holding capacity $\kappa$. A full description of this implementation is given in Speich et al. (2018b). The underlying equation is:

$$\frac{\gamma_r \times D_r}{L_r} = w_{ph} \times f_{seas} \times \frac{d\langle T \rangle}{dZ_e}, \tag{13}$$

where $\gamma_r$ is root respiration rate [mg C g$^{-1}$ roots day$^{-1}$], $D_r$ root length density [cm roots cm$^{-3}$ soil], $L_r$ specific root length [cm roots g$^{-1}$ roots], $w_{ph}$ photosynthetic water use efficiency [g C cm$^{-3}$ H$_2$O], $f_{seas}$ growing season length [fraction of a year] and $\langle T \rangle$ mean daily transpiration [mm day$^{-1}$] during the growing season. The left hand side represents the marginal cost of deeper roots, and the right hand side the marginal benefits, and solving for $Z_e$ gives the optimal rooting depth. Any equation can be used to relate $d\langle T \rangle$ to $dZ_e$. In this implementation, the probabilistic models of Milly (1993) and Porporato et al. (2004) are used for the understory and overstory, respectively. These two models reflect the differing water uptake strategies of grasses and trees (Guswa, 2010). Both models estimate transpiration based on soil water holding capacity $\kappa$ and long-term averages of climatic indices. Evaporative demand is represented by potential transpiration, and rainfall is represented as a marked Poisson process characterized by the frequency ($\lambda$ [events day$^{-1}$]) and mean intensity ($\alpha$ [mm event$^{-1}$]) of events. In FORHYCS, these variables are calculated as rolling means with a window of 30 years, including only the growing seasons.





Potential transpiration for the understory and overstory are taken from the calculations of the local water balance module (Sect. 2.1.4), and the rainfall characteristics are taken from modeled effective precipitation (i.e. after accounting for interception). The start and end of the growing season are determined based on the phenology module (Sect. 2.1.2). In addition, mean daily air temperature is calculated over the growing season to adjust respiration rate. The plant-specific parameters in Eq. 13 are

summarized in the variable $PP_o$ (see Speich et al., 2018b). A higher value of $PP_o$ indicates a greater difficulty for the plant to develop additional roots. In FORHYCS, $PP_o$ was set to $1.263 \times 10^{-4}$ for conifers and $1.01 \times 10^{-4}$ for broadleaved species. At cell level, $PP_o$ was averaged based on the relative share of aboveground biomass belonging to conifers and broadleaves. For the understory, the corresponding parameter $PP_u$ is set to $1.512 \times 10^{-4}$.

### 2.1.6 Snow-cover induced seedling mortality

For seedlings of the high-mountain species *Larix decidua*, *Pinus cembra* and *P. montana*, the model also includes the effect of snow-induced fungal infections via the variable $FDSA$ (final day of snow ablation), as described by Zurbriggen et al. (2014). An additional mortality term is calculated:

$$\mu_s = a \times FDSA^2 + b \times FDSA + c, \tag{14}$$

where $a$, $b$ and $c$ are empirical parameters fitted by Zurbriggen et al. (2014) for *Larix decidua* and for the two aforementioned

*Pinus* species. If $\mu_s$ is greater than background mortality or the mortality term integrating light, temperature and water stress, it is applied instead for the seedlings of these species. This aspect of the model was implemented to examine the feedback between forest dynamics and avalanches on a small spatial scale (Zurbriggen et al., 2014), but was never tested on landscape scale. Here, FDSA is defined as the last day of the year with more than 5 mm snow water equivalent.

### 2.1.7 Uncoupled mode and one-way coupling

The methods have so far described the model FORHYCS in its fully coupled version. It is also possible to run FORHYCS in uncoupled mode (without any information transfer between the hydrological and forest models), or with a one-way coupling (information transfer from the forest model to the hydrological model only). An uncoupled FORHYCS run consists essentially of a PREVAH run and a TreeMig run, happening independently from each other. Uncoupled FORHYCS differs from other PREVAH implementations mainly through the parameterization of soil and surface properties. As mentioned in Sect.

4.2, previous applications of PREVAH in Switzerland have used soil depth and water holding capacity from the agricultural suitability map BEK (BfR, 1980). Preliminary analyses in this project have shown that this parameterization gave implausible results when used with the newly implemented water balance module (Sect. 2.1.4). Therefore, to ensure comparability between coupled and uncoupled runs, all FORHYCS runs use the soil parameterization from Remund and Augustin (2015) (see Sect. 2.2.2) in forested cells. In uncoupled runs, it is assumed that the rooting depth of forests is 1 m. As this dataset was developed

based on forest soil profiles, values for cells outside currently forested areas are not reliable (Jan Remund, Meteotest, pers. comm.). To simulate forest expansion under climate and land use change scenarios, it was nevertheless assumed that cell val-





ues of the RA2015 (Remund and Augustin, 2015) dataset represent the water storage capacity for 1 m of soil depth. To account for shallower rooting of non-forest vegetation types, a land cover-dependent rooting depth parameter was introduced. The parameter values for different land cover types are given in Table S6. Non-vegetated land cover classes (e.g. built-up, bare rocks) use the same standard soil parameters as in original PREVAH (Gurtz et al., 1999). Another difference between FORHYCS

and PREVAH is the parameterization of canopy resistance. Whereas PREVAH gives a minimum canopy resistance for each land cover class (i.e. normalized by leaf area index), the new water balance module requires a minimum stomatal resistance. Following Guan and Wilson (2009), minimum stomatal resistance was set to 180 s m$^{-1}$ for forests, 130 s m$^{-1}$ for meadows and grasslands, and 210 m$^{-1}$ for shrubs.

One-way coupling is similar to the modeling experiment of Schattan et al. (2013): vegetation variables from TreeMig are
passed to PREVAH, but there is no feedback from the hydrological to the forest model. This configuration uses the abiotic drought index calculated with FORCLIM-E (Bugmann and Cramer, 1998). In this study, TreeMig was run with two soil datasets (BEK and RA2015; see below). In all cases, the hydrological part of the model uses the RA2015 dataset. In this study, rooting zone storage capacity SFC of the hydrological model was kept constant in one-way coupled mode, assuming a rooting depth of 1 m. Enabling climate-dependent adaptation of SFC in this mode would impact simulation results for the hydrology
part, but not for the forest.

## 2.2   The Navizence case study

### 2.2.1   Catchment description

The Navizence catchment is located in the central Swiss Alps and covers an area of 255 km$^2$. To enable the future migration of tree species that are currently not represented in the catchment, the modeling area extends beyond the catchment to form the
rectangular area shown on Fig. 2 (1079 km$^2$). The catchment is characterized by a sharp elevational gradient, with elevations ranging from 522 to 4505 m asl. Like in its neighboring valleys, this gradient is reflected in the hydro-climatic conditions. Due to the shielding effect of mountain ranges, the Rhône valley, where the catchment outlet is located, is the driest region of Switzerland, with mean annual precipitation (MAP; 1981-2010) at Sion totaling 603 mm (MeteoSwiss, 2014). However, the valley presents a strong altitudinal precipitation gradient, with MAP exceeding 2500 mm at 3000 m asl, most of it falling as
snow (Reynard et al., 2014).

Tree species composition shows a rather clear altitudinal zonation, with drought-resistant species (*Pinus sylvestris* and *Quercus spp.*) dominating at lower elevations, whereas *Picea abies, Larix decidua and Abies alba* dominate the subalpine stage and the treeline is formed by *Larix decidua* and *Pinus cembra*. The landscape is heavily influenced by human activity, with a large fraction of land occupied by settlements, cropland, vineyards and pastures. Furthermore, nearly all forests in the area are
subject to management in various forms and degrees of intensity, with a considerable impact on forest dynamics. Specifically, many forests were clear-cut between the Middle Ages and the nineteenth century, mostly for fuel (Burga, 1988). In the first half of the twentieth century, the quantitatively most important anthropogenic disturbance factors were litter collecting and wood pasture by goats (Gimmi et al., 2008). Nowadays, these practices have been largely abandoned, and timber harvesting plays





a limited role in the region. As a result of past and current anthropogenic factors, the main deviations from potential natural forest composition are (1) silvicultural practices favoring certain species, such as *Pinus sylvestris* and *Larix decidua* (2) Effects of litter removal and grazing and (3) a replacement of *L. decidua* and *Pinus cembra* by mountain pastures (Büntgen et al., 2006) and dwarf shrubs (Burga, 1988) near the treeline.

**Figure 2.** Map of the study area, including the Navizence catchment and the subcatchments used in this study. Streamflow measurements are available for the subcatchments 2 through 5. As the lowest part of the Navizence catchment (subcatchment 1) differs greatly from the others in terms of elevation and current land cover, modeled streamflow is also examined for this subcatchment, although no measurements are available. Forest-related model outputs are evaluated over the entire currently forested area, inside and outside the catchment. Streams are drawn for reference only.





Currently, three major hydropower plants are operational in the valley, with a total installed capacity of 164 MW and a mean net annual production of 570 GWh. The main reservoir is the artificial lake Lac de Moiry, located in a lateral valley, with a storage capacity of 77 mio m$^3$. A system of pipelines has been built to divert water from the Navizence, as well as from a neighboring catchment, into the lake.

### 2.2.2 Input data

Three kinds of spatial data are needed to run the model: daily meteorological data, time-invariant physiographic data, and spatially distributed model parameters for PREVAH. The model is driven by daily values for precipitation [mm], air temperature [°C], global radiation [W m$^{-2}$], wind speed [m s$^{-1}$], relative air humidity [%] and sunshine duration SSD [h], provided by the Swiss Meteorological Office MeteoSwiss (Begert et al., 2005).

Physiographic data consists of information on soil, topography and land cover. Soil is represented in terms of water holding capacity $\kappa$ [mm water depth mm$^{-1}$ soil depth] and soil depth. Two different datasets are used for soil properties. In previous applications in Switzerland, both PREVAH and TreeMig used grids of $\kappa$ and soil depth from a country-wide agricultural suitability map (BfR, 1980; hereafter referred to as "BEK"). The resulting rooting zone storage capacities (SFC [mm]) in the forested cells of the study region range between 3.75 and 110 mm. As this dataset was not specifically developed for use in forests, and some values are implausibly low, Remund and Augustin (2015) generated a new country-wide dataset (RA2015) of rooting zone storage capacity, on the basis 1234 forest soil profiles throughout Switzerland, combined with a lithological map. This dataset gives the volume of water that can be stored in the soil for a depth of up to 1 m, with lower values in cells where soil is assumed to be shallower. SFC in the new dataset ranges from 71 to 223 mm in forested cells of the study region. Figure S1 shows the rooting zone storage capacity in forested cells of the study region for both soil parameterizations. For coupled simulations, only the RA2015 parameterization is used. To facilitate comparison with results from previous studies, the parent models PREVAH and TreeMig are also run with the BEK parameterization.

### 2.2.3 Comparison data and metrics of agreement

This section describes the data against which model outputs were compared, including three datasets of vegetation properties and one dataset of streamflow measurements. These datasets are used to plausibilize model outputs and serve as a basis for the choice of model configuration.

The simulated stem numbers and above-ground biomass were compared against data from the first Swiss National Forest Inventory (NFI; Bachofen et al., 1988). As the sampling plots of the NFI are distributed on a regular grid, each plot is a randomly selected from all forest plots in that region, and may not be considered representative for a larger area. Therefore, the 245 NFI plots in the study area were aggregated to seven classes based on aspect and elevation, with four elevation bands for North-facing plots and three for South-facing plots. This way, each class has a sample size of at least 30 plots, which ensures that the averages are representative. For comparison with inventory data, simulated biomass was also averaged over the same strata.





Simulated LAI was compared to the remotely sensed LAI dataset provided by Copernicus at 300 m resolution (Copernicus Service Information, 2017). This dataset uses measurements of the satellite Proba-V and its temporal coverage starts in January 2014. The LAI 300m rasters were resampled to match the resolution and extent of the model input and output, then stratified as described above. For each cell of the original LAI 300m grids, the maximum value of the 10-day periods contained

between May and July of the years 2014 through 2016 was used as an estimate of maximum LAI, independent of intra-annual fluctuations. As forest properties are also shaped by local processes not represented in the model (e.g. disturbance, forest management), a direct cell-by-cell correspondence of simulated and observed values is not expected. Furthermore, cells of the remote sensing dataset are spatially heterogeneous and may include non-forested parts. However, it was still assumed that over larger domains, a good correspondence between simulated and observed LAI would be indicative of good model performance.

Therefore, the simulations and observations were also stratified by elevation and aspect. As the number of cells is relatively large, smaller elevation bands were chosen than for inventory data. The cells are divided between North and South facing aspects, then binned into nine elevation bands. These elevation bands have a fixed width of 200 m, except the lowest (< 700 m asl) and highest (> 2100 m asl). Each zone contains between 100 and 714 forested cells. Model outputs and observations are only evaluated over the areas classified as forests or shrubland in the model input.

The recently developed Switzerland-wide vegetation height dataset at 1 m resolution of Ginzler and Hobi (2016), derived from stereo aerial images, was compared against simulated canopy structure. To compare the level of agreement between observed and simulated canopy structure, two novel metrics were introduced. The rationale behind these metrics is illustrated in Figure 3. Both measures are based on the discrete height classes used in TreeMig, and use the cumulative fractional cover of each height class, starting from the highest class. In the illustrative example of Fig. reffig:fh3 a), the cumulative fractional

cover of the height class "5-10 m" corresponds to the visible crown area of the height classes "5-10 m" and "10-15 m", divided by the potentially vegetated area (i.e. excluding the area occupied by a road, shown in grey). In each cell of the model grid, the fractional cover of each height class is calculated from simulated species-size distribution using the procedure described in Eqs. 3 and 4. As can be seen on the examples in Fig. 3 c) and d), each cell from the model grid covers 200x200 cells of the observations grid. Therefore, for each model cell, the observed fractional cover is calculated from the relative number of

high-resolution cells belonging to each height class. Observed cells with a height of zero represent non-vegetated surfaces such as roads, water bodies or buildings and were excluded from the analysis. Indeed, as the model does not contain any land-cover information on subgrid level, these elements are an irreducible source of disagreement between observations and simulations. On the other hand, observation cells with a height between zero and 1.37 m are assumed to be covered by ground vegetation, and decrease the total fractional canopy cover in a coarse cell. The first cell-level measure of agreement is the difference in

observed and simulated H95, i.e. the lowest height class for which the cumulated fractional cover (starting from zero) equals or exceeds 95% of the total fractional cover. Figure 3 b) shows the H95 of the observations, at the level of model cells. The second measure of agreement is illustrated in Fig. 3 e) and f). The cumulative fractional cover of each class (starting from the top) is plotted for observations and simulations. The better the agreement, the closer the curves are to each other. Therefore, the second measure of agreement, termed 1-ABC (where ABC stands for "Area Between the Curves"), is defined as the fraction of the plot

area not contained between the curves. The plot in Fig. 3 e) applies this to the cell shown in Fig. 3 c), and represents a case with





a poor agreement between simulations and observations. Indeed, this area was devastated by a wildfire, and is thus currently very sparsely forested. As this fire is not represented in the model, the simulations indicate a fully developed forest. Even in this extreme case, the ABC does not exceed 40% of the plot area. Therefore, a 1-ABC score of 0.6 can be considered a poor fit. On the other hand, the sample cell in Fig. 3 d) and f) shows a good agreement between observations and simulations, with

1-ABC exceeding 0.99. These two measures of agreement can be used to evaluate the performance of a model by examining their distribution over the whole simulation domain. A better performing model will have a higher proportion of cells with a dH95 close to zero and a 1-ABC close to one. Furthermore, the spatial distribution of dH95 and 1-ABC may give insight into the factors that contribute to agreement or disagreement between simulations and observations.

### 2.2.4    Simulation experiments

To evaluate the behavior of the coupled model FORHYCS and the importance of the different forest-hydrology couplings implemented, two series of simulation experiments have been conducted. An overview of the different simulation runs is given in Table 1. In the first series of experiments, the simulations start with no forest, and a full succession is modeled. The simulations span a period of 515 years, where the last 45 years are the years 1971 to 2015. For the first 470 years, the meteorological forcing consists of years bootstrapped from the period 1981 to 2000. In two cases (Succ_TM_BEK and

Succ_TM_RA2015), the model is run in uncoupled mode, and only the forest output is evaluated. This is equivalent to a standard TreeMig run. The difference between the two runs is the parameterization of the rooting zone storage capacity for the (abiotic) drought stress module (FORCLIM-E; Bugmann and Cramer, 1998). In the first case, the storage capacity in each cell is given by the soil depth and water holding capacity given in the Swiss soil map for agricultural suitability (BEK; BfR, 1980), as in previous TreeMig applications in Switzerland (e.g. Bugmann et al., 2014). In the second case, the parameterization of

Remund and Augustin (RA2015; 2015) is used. As noted in Sect. 2.2.2, the soil water holding capacity is much larger in the RA2015 dataset for most cells. As a result, the (abiotic) drought index also shows great differences between the two model runs. Figure S1 c) and d) shows the difference in mean annual drought index (1971-2015), and maximum annual drought index between the BEK and RA2015 parameterizations. Due to the considerable effect of maximum height reduction (Sect. 2.1.3) on the coupled model, two additional TreeMig runs were performed with this effect enabled, to facilitate the comparison between

coupled and uncoupled runs.

In coupled mode, FORHYCS is run with different configurations, with the various couplings described in Sect. 2.1 switched on or off (maximum height reduction, stress-induced leaf area reduction, dynamically varying rooting depth, and snow-induced seedling mortality). Based on pilot study results, the configuration Succ_noLAred (all couplings switched on, except leaf area reduction) was selected as the best configuration, and the other configurations in Table 1 differ from Succ_noLAred by only

one process switched on or off. The configuration Succ_noLAred is also used for the second set of model runs, which starts in the year 1971 and ends in 2100. In this second set of model runs, the sensitivity to a ramp-shaped climate change is evaluated (see Fig. 4). In the period 1971-2015, observed forcing is used. From 2016 to 2100, years are randomly selected from the period 1981-2015 (excluding the abnormally dry and hot year 2003). Furthermore, from 2016 on, daily temperature is incremented by a given amount of degrees dT, and daily precipitation is scaled by a given factor dP. The values of these factors are given in



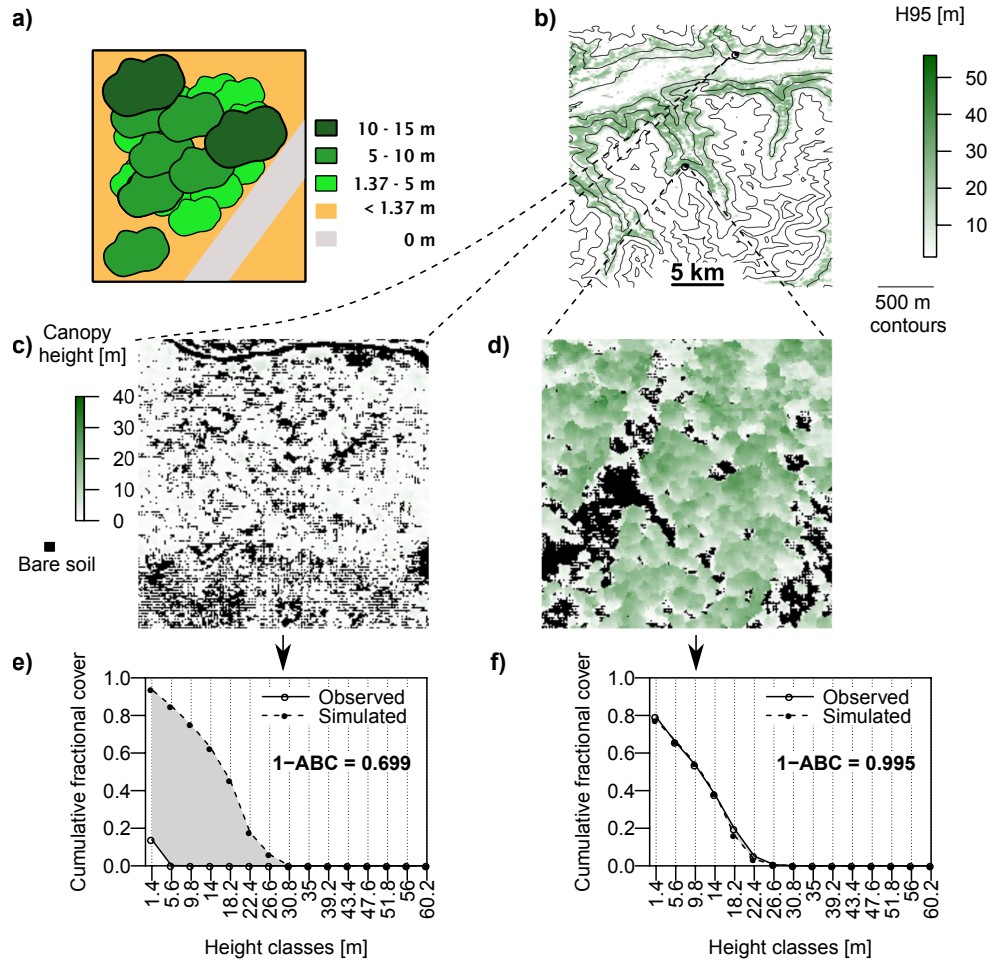

**Figure 3.** (a) Schematic representation of the canopy division into discrete height classes, as used in the model-data comparison. The cumulative fractional cover of a height class is the total visible crown area of that and higher classes, divided by the total vegetated area. In this example, a road (in grey) crosses the cell, which reduces the total vegetated area. (b) For each 200x200 m cell, the shade of green represents the lowest height class for which 95% of the 1x1 m cells are lower or equal (H95). (c) An example of a very sparsely vegetated 200x200 m cell. The shade of green shows the height of each 1x1 cell. Cells with a height of zero are assumed to be bare and are marked black. This cell is located in an area disturbed by a wildfire in 2003. (d) An example of a mountain forest, located at 2000 m asl. The bare areas in this cell are mostly covered by rocks. (e) and (f) An illustration of the 1-ABC index of agreement between observed and simulated forest structure, applied to the cells shown in c) and d). The open dots and solid lines show the cumulative sum of 1x1 cells belonging to each discrete height class, starting from the right (i.e. from the highest class), normalized by the area of the 200x200 m cell which is not bare. The full dots and dashed lines represent the cumulative relative coverage of each height class as simulated by FORHYCS. As the wildfire is not reflected in the model, the simulation shows a fully developed forest in the cell shown in c), leading to a poor match between simulated and observed canopy structure. On the other hand, the forest simulated in the cell shown in d) corresponds well to the observed structure, leading to a high 1-ABC score.



**Table 1.** Overview of the conducted simulation experiments

| Simulation name | Description | Years |
|---|---|---|
| Succ_TM_BEK | Full succession with uncoupled TreeMig, soil AWC from BfR (1980) | 470 years bootstrapped from 1981 to 2000, followed by 1971-2015 |
| Succ_TM_RA2015 | Full succession with uncoupled TreeMig, soil AWC from Remund and Augustin (2015) | idem |
| Succ_Full | Full succession, with all forest-hydrology couplings enabled | idem |
| Succ_noLAred | Full succession, without stress-induced reduction of LAI | idem |
| Succ_cSFC | Like Succ_noLAred, with constant SFC (assuming 1 m rooting zone depth) | idem |
| Succ_noHmax | Like Succ_noLAred, without drought-induced height limitation | idem |
| Succ_noHmax | Like Succ_noLAred, without snow-induced seedling mortality | idem |
| Clim_11 | Future simulations, with a temperature increase of $x$ K and a precipitation change of factor $y$. | Years 1971-2015 with observed meteorological forcing, then 2016-2100 with bootstrapped years and modified T and P. |
| Clim_00 | Idem, but without vegetation dynamics (hydrology only, default parameters) | idem |
| Clim_01_BEK | Idem, but with one-way coupling (TreeMig parameterized with BEK soil) | idem |
| Clim_01_RA2015 | Idem, but with one-way coupling (TreeMig parameterized with RA2015 soil) | idem |
| Clim_11_NCS | Future simulations, without considering the effect of $CO_2$ on stomatal resistance | idem |
| Clim_11_LC | Future simulations in which forest is allowed to grow in all potentially forested cells | idem |





**Figure 4.** Workflow of the various simulation experiments conducted in this study. The Succession runs (Succ_) use bootstrapped meteorological forcing for 470 years, followed with observed forcing for the period 1971-2015. FORHYCS is run with several configurations, as described in Table 1. The output from each run is then compared against observations, based on which one configuration is selected for the runs under idealized climate change (Clim_). The modifiers for temperature and precipitation (red line) are scaled linearly between zero and their maximum in the period 2016-2050. Atmospheric $CO_2$ concentration $C_a$ (blue line, approximate illustration) has no effect on the temperature and precipitation modifiers, but impacts the canopy resistance.





Table 1. To emulate a gradual progression of climate change, these factors are scaled linearly between zero and their full value between 2016 and 2050. The runs Clim_11_NCS test the impact of the $CO_2$ effect on stomatal closing, implemented through Eq. 12. In these runs, the $CO_2$ response function is always set to one, i.e. stomatal response to high $CO_2$ is switched off. In all the runs presented so far, forest growth is restricted to the currently forested cells. In the runs Clim_11_LC, forest is allowed

to grow in all potentially forested land cover classes. Thus, the potential ecohydrological consequences of land abandonment and rising treelines are examined.

## 3   Results

### 3.1   Plausibilization of simulated streamflow

To evaluate model efficiency, the Kling-Gupta efficiency (Gupta et al., 2009, KGE,) was applied to daily streamflow for the

period April 2004-December 2008 in subcatchments 2 to 5 (Table 2). The scores were calculated for three different model runs: uncoupled FORHYCS (Clim_00), fully coupled FORHYCS (Clim_11), as well as a run with original PREVAH (the version used in Speich et al., 2015) for reference. There is little difference between the scores of these three runs, and no model version consistently outperforms the others. The last four columns of Table 2 show the observed and simulated mean annual streamflow for the period 2005-2007 (the years for which there are no gaps in the observations). The sums simulated

by PREVAH are consistently greater than for FORHYCS, with differences between PREVAH and uncoupled FORHYCS ranging between 40 (Moiry) and 172 mm year$^{-1}$ (Chippis). The values simulated with coupled FORHYCS are somewhat higher than with the uncoupled version at the lower elevation subcatchments (35 mm year$^{-1}$ in subcatchment 1 and 6 mm year$^{-1}$ in subcatchment 2), but almost equal in the two high-elevation catchments 4 and 5. Figure 5 a) shows the daily values (30-day rolling means) for subcatchment 3 (Vissoie; analogous figures for the other gauged subcatchments are given on Fig.

S2-S4). The main differences between PREVAH and the FORHYCS runs occur in late summer and autumn, where streamflow simulated by PREVAH is consistently higher. The differences between the two FORHYCS versions are shown on Fig. 5 b). The greatest differences occur in winter and early spring, with some peaks in spring 2005 an 2006, and consistently higher streamflow in the winters 2006-2007 and 2007-2008.

### 3.2   Forest spin-up with different model configurations

### 3.2.1   Biomass and species composition

Figure 6 shows the aboveground biomass simulated with FORHYCS using the configuration Succ_noLAred (analogous figures for the other configurations are provided in S2; Fig. S5 to S12). In addition, the bar shows the average aboveground biomass of trees in plots of the first Swiss national forest inventory (1982-1986; Bachofen et al., 1988). The simulations start in 1500 with no trees. Biomass increases quickly at the beginning, so that in most zones, values close to 100 t ha$^{-1}$ are reached within

the first 40 years. The initial species composition consists of various broadleaf species, among which the maple species *Acer campestre* and *A. pseudoplatanus*. The former is more prevalent at lower elevations, and the latter at higher elevations. After





**Table 2.** KGE scores obtained by three different model configurations against daily observed streamflow data for the period 2004-2008 (first three columns), and mean annual streamflow sums $Q_a$ [mm year$^{-1}$] for the period 2005-2007 (observed and simulated).

| Subcatchment | KGE PRE-VAH | KGE Clim_00 | KGE Clim_11 | $Q_a$ Obs | $Q_a$ PR | $Q_a$ FH_00 | $Q_a$ FH_11 |
|---|---|---|---|---|---|---|---|
| 1. Chippis | - | - | - | - | 621 | 449 | 484 |
| 2. Moulin | 0.74 | 0.72 | 0.73 | 735 | 710 | 614 | 620 |
| 3. Vissoie | 0.66 | 0.71 | 0.7 | 568 | 664 | 542 | 553 |
| 4. Mottec | 0.83 | 0.8 | 0.8 | 1211 | 1189 | 1140 | 1141 |
| 5. Moiry | 0.84 | 0.87 | 0.87 | 823 | 957 | 917 | 917 |

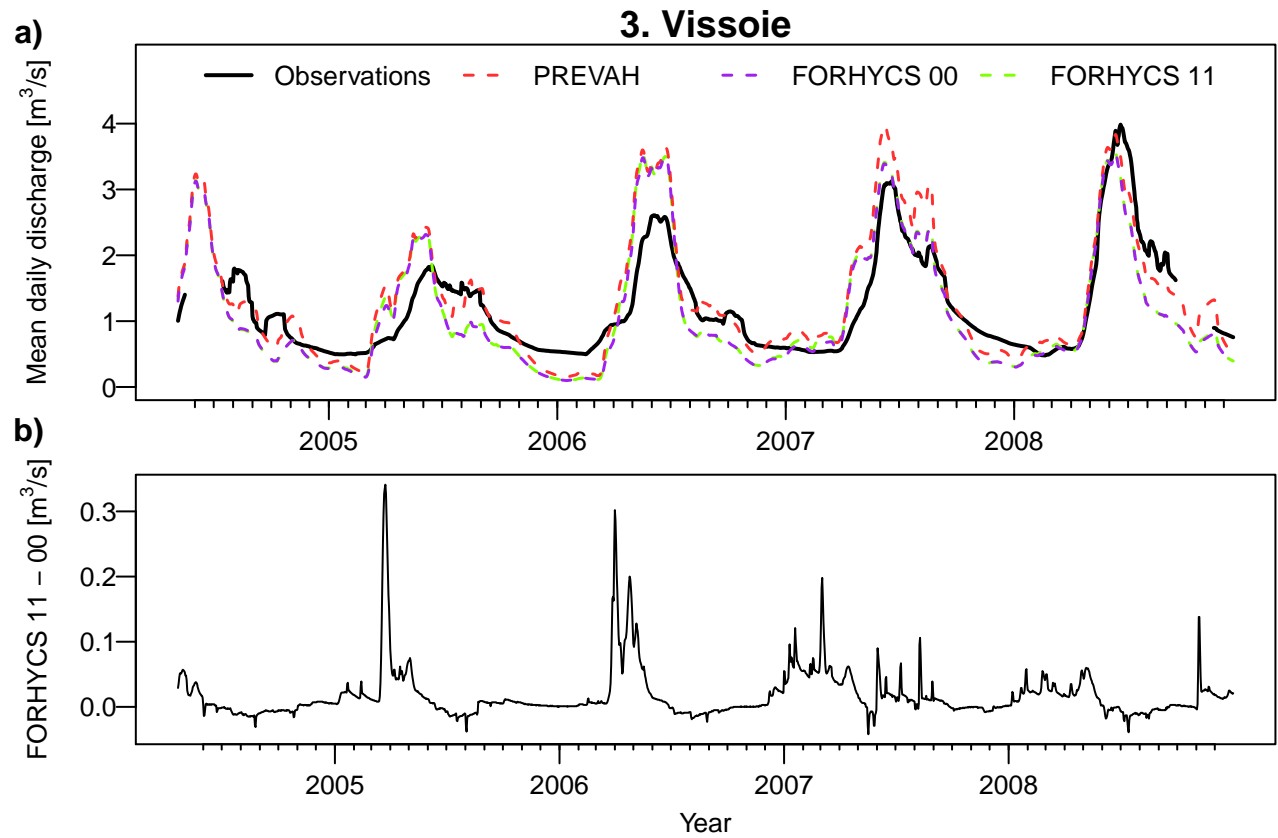

**Figure 5.** (a) Observed vs. simulated daily streamflow for subcatchment 3 (Vissoie) for the period 2004-2008. For clarity, the plot shows rolling averages with a 30-day window. The plot shows results for two FORHYCS runs, uncoupled (FORHYCS 00) and fully coupled (FORHYCS 11). For reference, the results obtained with standard PREVAH are also shown. (b) Difference in simulated daily streamflow between the coupled and uncoupled versions of FORHYCS. Streamflow simulated with coupled FORHYCS is usually higher than for the uncoupled version, and the greatest differences occur in winter and spring.





this initial state, *Pinus sylvestris* (mainly at lower elevations) and *Larix decidua* (mainly at higher elevations) start developing. Finally, *Quercus* species (lower elevations, mainly *Q. pubescens* and *Q. petraea*), *Abies alba* (north-facing slopes) and *Picea abies* develop, with the latter becoming dominant at higher elevations. At the end of the simulation, the forest seems to have reached a state in which the relative importance of species changes little over time. At lower elevations, the state at the end

of the simulation is close to that reached after the first 100 or 200 years. By contrast, at high elevations, the replacement of *L. decidua* by *P. abies* occurs over a much longer time.

At low elevations, simulated biomass is higher than the inventory value, especially on north-facing slopes. By contrast, simulated biomass is lower than the observed value at intermediate elevations. Unlike in simulations, for slopes reaching below 1346 m, there is almost no *Quercus* and *Acer* biomass in the inventory data, and the share of other broadleaves is also much

smaller. Also at intermediate elevations, FORHYCS simulates a higher biomass for *Acer* and other broadleaves. However, *L. decidua* makes up one third of observed biomass in that elevation band, while it is practically non-existent in the simulations. Also at higher elevations, the simulated biomass of *L. decidua* is much lower than in the inventory data.

The results of Succ_TM_BEK_noHmax (Fig. S5) differ greatly from those on Fig. 6. *Pinus sylvestris* is dominant at intermediate elevations, whereas high elevations are dominated by *L. decidua*. *P. abies* is hardly represented, and simulated biomass

for broadleaves is very low. Fluctuations of simulated biomass are also much higher, with increases or decreases of up to 70 t ha$^{-1}$ within 20 years. For TM_RA2015_noHmax (Fig. S6), biomass equals or exceeds 200 t ha$^{-1}$ in all strata except at the highest elevations. The mesophilous species *Abies alba* and *P. abies* dominate at all elevations. For TreeMig runs with maximum height reduction (TM_BEK and TM_RA2015; Figs. S7 and S8), species composition is similar to the results of standard TreeMig, but with much lower biomass.

The configuration Succ_Full differs from Succ_noLAred in that LAI is reduced as a function of drought or low temperatures. Biomass simulated with this configuration (Fig. S9) is markedly higher than with Succ_noLAred, especially at lower and intermediate elevations, while species composition is similar. Biomass also fluctuates more for Succ_Full at lower and intermediate elevations. For Succ_noHmax (Fig. S10) and Succ_cSFC (Fig. S11), biomass is also higher than for Succ_noLAred. In the former case, species composition is similar to Succ_noLAred, whereas for Succ_cSFC, the share of relatively drought intolerant

species is larger at lower elevations (e.g. higher share of *P. abies* and lower share of *Pinus sylvestris*). Biomass simulated with Succ_noSmort (Fig. S12) shows no apparent difference from Succ_noLAred.

### 3.2.2 Canopy structure

The distribution of the two metrics of agreement between observed and simulated canopy structure, is shown on Fig. 7. The distribution of $\Delta H95$ shows large differences between the model configurations. While Succ_TM_RA2015_noHmax has its

maximum count of $\Delta H95$ at 25.2 m and overestimates H95 in almost all cells, Succ_TM_BEK_noHmax shows a much flatter distribution, with a large number of under- and overestimations. By contrast, the TreeMig runs with maximum height limitation have their maximum count at 0 m, suggesting a better fit. Most of the FORHYCS runs show a similar pattern, with a peak close to 0 m, and most values contained between -16.8 and 21 m. Succ_noHmax, however, has its maximum at 16.8 and overestimates H95 in almost all cells. The distribution of the 1-ABC scores, shown in the lower plot, also sets apart the







**Figure 6.** Aboveground tree biomass simulated with FORHYCS using the configuration Succ_noLAred (Table 1). The graphs show annual values, averaged over seven clusters of cells. The bar shows the aboveground biomass in the same area, from the first Swiss national forest inventory (1982-1986; Bachofen et al., 1988). The limits of the elevation bands were set so that each cluster contains at least 30 forest inventory plots. The dashed line marks the year 1971, from when meteorological data are available. Simulation years before 1971 use meteorological data bootstrapped from the years 1981-2000.





two configurations with the largest overestimation of H95, Succ_TM_RA2015_noHmax and Succ_noHmax. These runs have their highest density at a lower value than the other configurations. The other FORHYCS runs all have their highest density around 0.95. While Succ_Full shows a distribution of dH95 that is very similar to Succ_noLAred and Succ_noSmort, its density distribution for 1-ABC differs from that of the other 2 configurations. Succ_Full has a lower density around 0.95, but

a higher density between 0.7 and 0.85, indicating a lower degree of agreement between observations and simulation for this configuration.

### 3.2.3    Leaf Area Index

Figure 8 shows a comparison of the Copernicus 300m LAI and values simulated with the different TreeMig and FORHYCS configurations listed in Table 1. For the observations, the averages for the different elevation and aspect classes range approx-

imately between 2 and 4. The lowest values occur on south-facing slopes below 700 m asl (i.e. close to the bottom of the Rhône valley). LAI of the south-facing slopes increases steadily with elevation up to 1700-1900 m asl, where it reaches a value of 4. For cells over 2100 m asl, LAI is again somewhat smaller. LAI is generally higher on north-facing slopes. There is little difference among the elevation bands between 700 and 2100 m asl. Average LAI is always around 4 in that elevational range. Smaller values occur only in the lowest and highest elevation bands. The two standard TreeMig runs differ greatly in

the range and pattern of simulated LAI. For Succ_TM_BEK, parameterized with available water capacity (AWC) from the soil suitability map (BfR, 1980), the values range between 2 and 6. The highest values occur in the lowest elevation band and, for north-facing slopes, at 1700-1900 m asl. Except for the lowest elevation band, LAI on north-facing slopes is markedly higher than on south-facing slopes, with differences of up to 2.5. By contrast, for Succ_TM_RA2015, which uses the AWC from Remund and Augustin (2015), the absolute values are much higher and the variability much lower. For all elevation bands,

values range between 6.5 and 7.5. The results for TreeMig runs with height limitation are almost equal to the standard version.

There is little difference in patterns and absolute values among the FORHYCS runs. The absolute values range from 4 to 7. Spread is lowest for the two highest elevation bands, where all configurations give a value of approximately 6, for both north- and south-facing slopes. At lower elevations, there is a clear difference between the two aspect classes, with consistently higher values for the north-facing slopes. The difference between configurations increases with decreasing elevation, and is somewhat

higher on south-facing slopes. The configuration Succ_cSFC consistently returns the largest values. On north-facing slopes, the values are lowest for Succ_noLAred. This is also the case at higher elevations on south-facing slopes, whereas at lower elevations, Succ_noHmax returns the lowest values. In most cases, the results of Succ_Full and Succ_noLAred are similar. Up to 1300-1500 m asl on north-facing slopes, Succ_Full gives somewhat higher values, whereas on south-facing slopes, the value for Succ_noLAred is higher. In all cases, the symbols for Succ_noSmort are indistinguishable from Succ_noLAred.





**Figure 7.** Distribution of the two goodness-of-fit metrics for canopy structure described in Sect. 2.2.3 over all forested cells of the simulation domain (n=7138). The upper graph shows the distribution of $\Delta H95$ (difference between H95 for simulated and observed data) for the different model configurations listed in Table 1. More values closer to zero indicate a better agreement between observed and simulated canopy structure. As H95 uses the height classes of TreeMig, the results are given as discrete values with an interval of 4.2 m. The lower graph shows the density distribution of the 1-ABC scores (bandwidth=0.0075). The better the agreement between observed and simulated canopy structure, the more values are close to one.



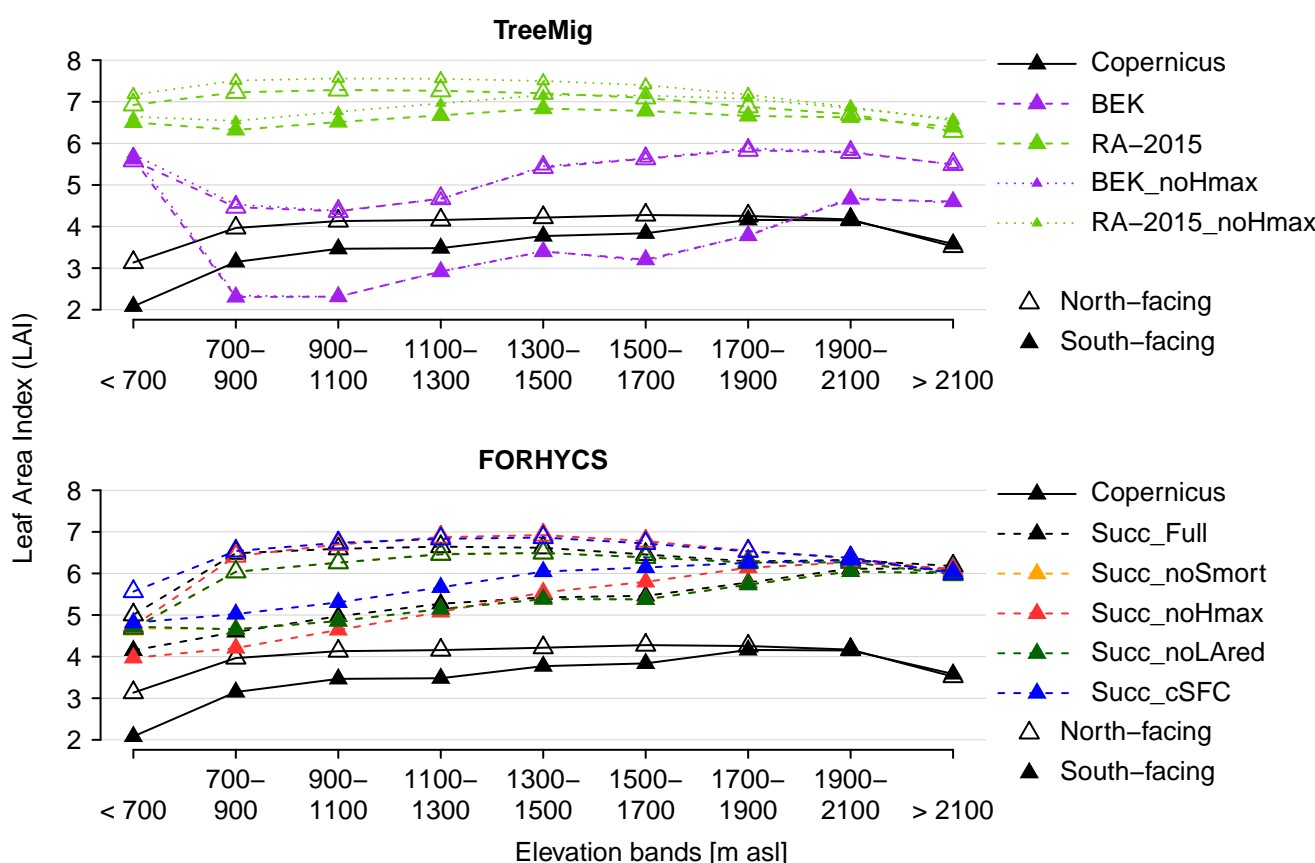

**Figure 8.** Observed and simulated Leaf Area Index (LAI), averaged over elevation bands and aspect classes. The observed values show the average of cell-level maximum LAI of the period 2014-2016. The simulated values correspond to the averages of the largest annual maximum LAI values in the last three years of the succession runs (2013-2015). The top graph shows LAI simulated with TreeMig, using two different parameterizations for soil moisture storage capacity. The purple symbols show results of TreeMig runs parameterized with the soil suitability map (BfR, 1980), whereas the green symbols correspond to TreeMig runs parameterized with the dataset of Remund and Augustin (2015). The bottom graph shows the results of the various FORHYCS succession runs, with the configurations listed in Table 1.



### 3.3 Idealized climate change runs

#### 3.3.1 Differences between uncoupled, one-way coupled and fully coupled model runs

Figure 9 shows how area-averaged LAI and rooting zone storage capacity SFC change for all idealized climate change scenarios (the results are shown here as rolling means with a 30-year window; a version without smoothing is given in Fig. S-13). LAI

and SFC are averaged over two of the strata used in Fig. 6, the lowest stratum of south-facing cells (Fig. 9 a) and c)) and the highest stratum of north-facing cells (Fig. 9 b) and d)). Fig. 9 a) shows the results of the coupled run for the low-elevation south-facing cells. At the end of the succession run, LAI is approximately 6, and over the next century, the values range between 3.5 and 6.5. Long-term average LAI is lower for warmer and drier scenarios. The plot of annual values (Fig. S-13) shows that under warmer scenarios, the variability of LAI is much higher, with LAI decreasing much more in certain years than

in less warm scenarios. In other years, annual LAI values strongly converge between scenarios. For SFC, values decrease for drying scenarios and increase for wetting scenarios. The high-elevation north-facing cells (Fig. 9 b), Fig. S-13 b)) also show a decrease in long-term average LAI and an increase in variability under drying and warming scenarios. SFC initially increases in all scenarios. Around 2050 (when the temperature and precipitation modifiers reach their maximum), SFC starts decreasing, with a faster decrease in drying scenarios.

Due to the similarity of LAI simulated by TreeMig with and without height reduction (cf. Fig. 8), only the results for the version without height reduction are shown. LAI differs greatly between the two TreeMig simulations, especially for the low-elevation south-facing cells (Fig. 9 c), Fig. S-13 c)). With TM_BEK, LAI is approximately at 3.5 at the end of the succession run, and decreases to less than 1.5 in the warmest and driest scenario (T6_P-10). By contrast, under TM_RA2015, LAI decreases from 6.5 to 5 in the T6_P-10 scenario. Inter-annual variability increases with TM_RA2015, but not with TM_BEK

(Fig. S-13 c)). In the high-elevation north-facing cells, the LAI trajectories diverge between the two TreeMig runs. Under TM_BEK, as for the lower elevation cells, LAI decreases under the T6_P-10 scenario while inter-annual variability increases. On the other hand, with TM_RA2015, LAI increases under all warming scenarios.

Figure 10 a) shows simulated annual streamflow (30-year rolling means) for subcatchment 1 (Chippis), as simulated with uncoupled FORHYCS. From initially 550 mm year$^{-1}$, streamflow decreases by about 50% under the most extreme warming

and drying scenario (temperature increase of 6K and precipitation decrease by 10 %) . Figure 10 b), c) and d) show the difference in annual streamflow to the uncoupled run (Fig. 10 a)), for the one-way coupled runs (TM_BEK and TM_RA2015) and the fully coupled FORHYCS runs, respectively. In all cases, streamflow is higher in the coupled runs. For TM_BEK, the difference is approximately 30 mm year$^{-1}$ at the end of the succession and increases under idealized climate change. The increase is greater for warmer scenarios, with a difference of 60 mm year-1 for the warmest scenarios (T6_Py) at the end of

the simulation. The difference in annual streamflow is much less for TM_RA2015, and ranges between 5 and 20 mm year$^{-1}$. Here, the difference in streamflow decreases with idealized climate change, with a more pronounced decrease for the warmer scenarios. For the fully coupled runs, the initial difference is around 30 mm year$^{-1}$, which is similar to the initial difference for TM_BEK. During idealized climate change, the difference does not increase as much as for TM_BEK: the largest difference

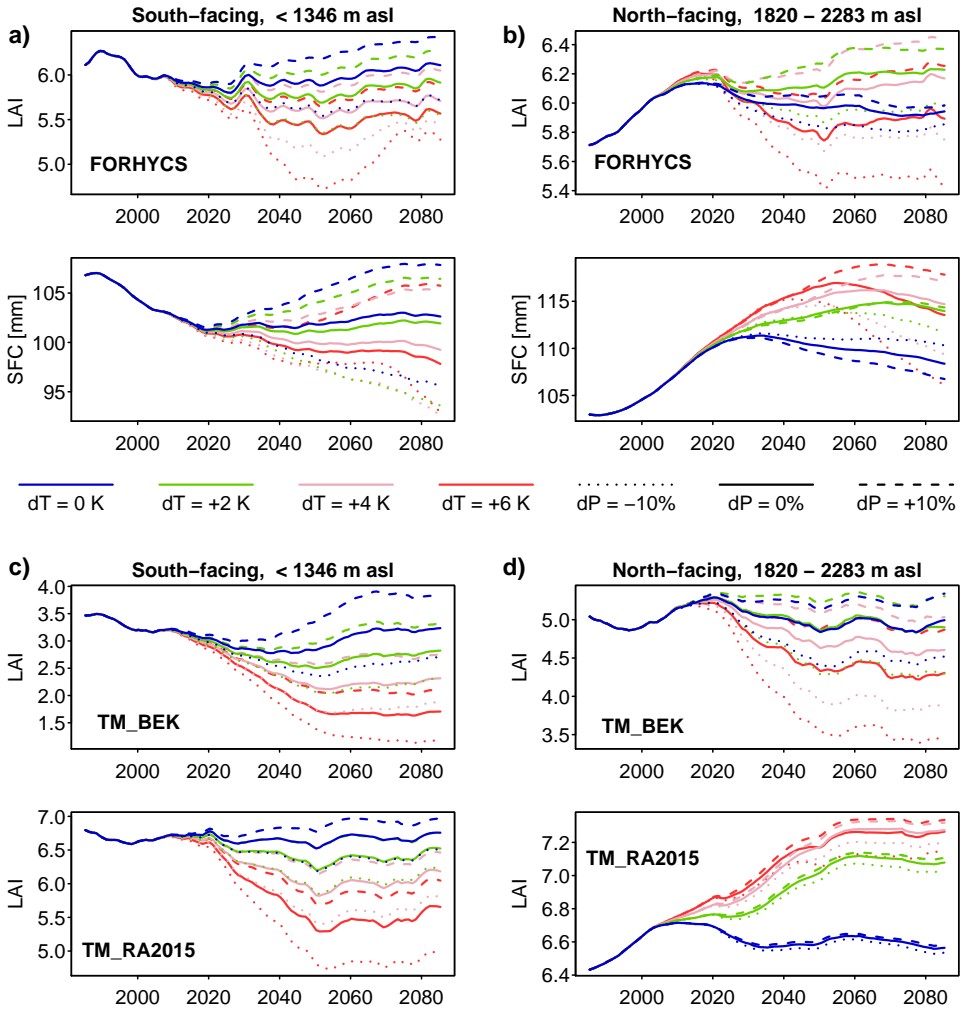

**Figure 9.** (a) Annual maximum LAI (top) and rooting zone storage SFC (bottom) under idealized climate change scenarios, simulated using coupled FORHYCS (Clim_11). LAI and SFC are averaged over the cells belonging to the lowest elevation class with south-facing slopes (same stratification as for Fig. 6). The results are shown as 30-year rolling means. Both LAI and SFC generally increase under wetting scenarios, and decrease under drying scenarios, although trends are not always monotonous. (b) Same as a), but for the highest elevation class with north-facing slopes. Also here, LAI increases under wet scenarios and decreases under dry scenarios. SFC trajectories show large differences between scenarios. (c) LAI simulated with TreeMig, with the two different soil parameterizations (Clim_01_BEK and Clim_01_RA2015). Note the different scales on the y-axis. LAI simulated with Clim_01_BEK is markedly lower than for Clim_11, whereas the Clim_01_RA2015 values are somewhat higher. (d) Same as c), but for the highest elevation class with north-facing slopes.





in the order of 40 mm year$^{-1}$. Unlike for TM_BEK, the scenarios with the greatest difference by the end of the simulation are the three warming and drying scenarios.

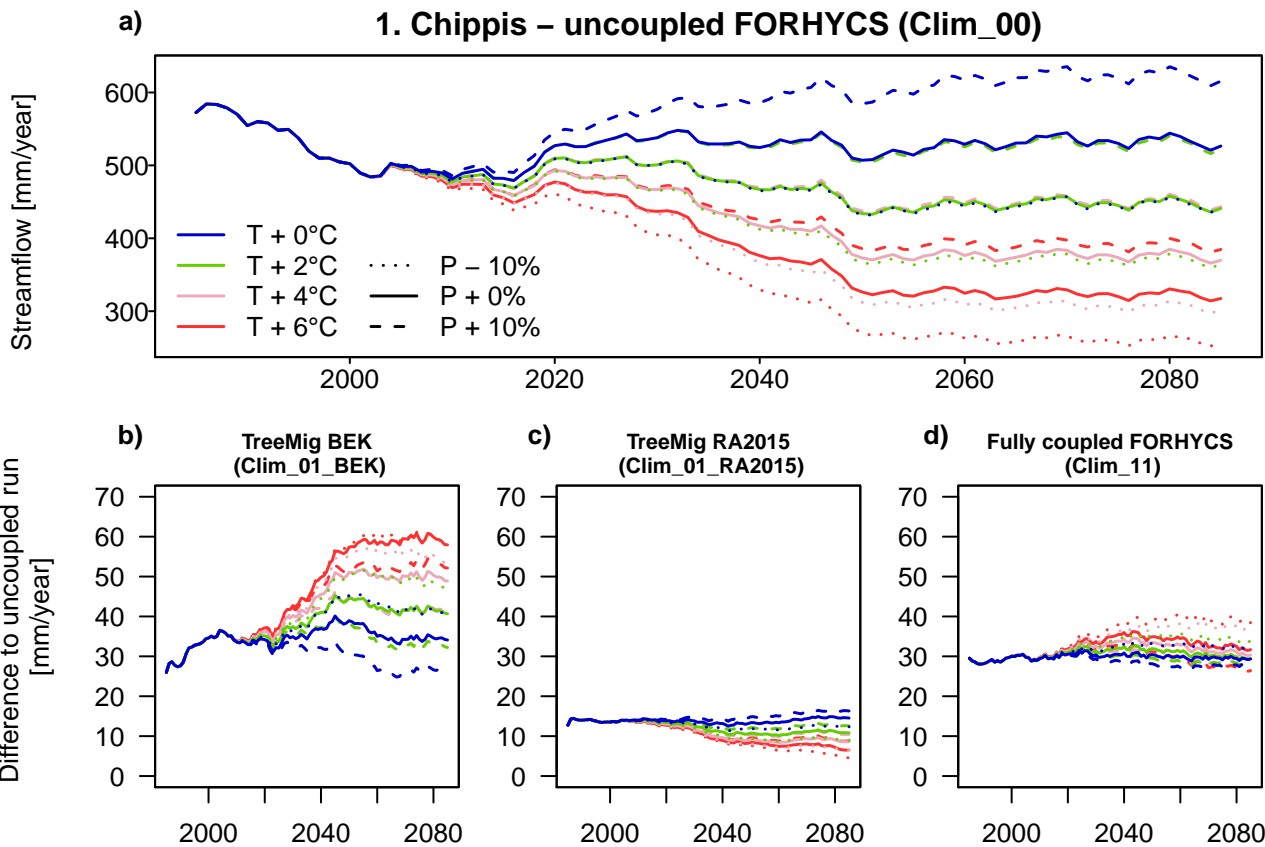

**Figure 10.** (a) Simulated annual streamflow (30-year rolling means) in the Chippis subcatchment (lowest elevation, most forested). In the warmest dry scenario (T6_P-10), annual streamflow is reduced approximately by half, from 550 to less than 300 mm/year. (b) and (c) Difference in annual streamflow in the runs with one-way coupling in the warm and dry scenarios, relative to the uncoupled run. (d) Difference in annual streamflow in the fully coupled FORHYCS run, relative to the uncoupled run. In all coupled runs, streamflow is greater than in the uncoupled version, due to lower LAI (see Fig. 8 and 9; the standard PREVAH value for forest LAI is 8) and, for the fully coupled version, smaller rooting zone storage capacity SFC

### 3.3.2 Effect of $CO_2$ concentration

The increase of stomatal resistance due to increased $CO_2$ concentration (Eq. 12) has a minimal effect on streamflow and area-averaged forest properties (not shown). In Clim_11_NCS (the runs in which Eq. 12 is set to one), streamflow is consistently lower than in the Clim_11 runs. The greatest difference occurs towards the end of the simulation period in the Chippis sub-catchment, with differences of up to 10 mm year$^{-1}$. The differences are largest in the simulations with a precipitation increase.



Regarding vegetation properties, the differences in LAI and SFC, averaged over the strata used in Fig. 6, never exceed 0.05 $m^2 m^{-2}$ and 1 mm, respectively.

### 3.3.3  Effect of land-use change

Figure 11 shows the difference in annual streamflow (30-year annual means) between a fully coupled run without land-use
5   change (Clim_11) and a run in which forest is allowed to grow in all cells with a "potentially forested" land cover type (see Fig. 2). The three subcatchments shown here differ by elevation and distribution of land cover classes. Chippis is the lowest subcatchment and has few cells belonging to the "potentially forested" land cover classes. By contrast, the high-elevation subcatchments Mottec and Moiry are barely forested (Mottec) or have no forest cells at all (Moiry). In all cases, allowing land-cover change leads to a decrease in streamflow. The magnitude of this change depends greatly upon the warming scenario.
10  When no warming is assumed, the difference in streamflow relative to the simulation without land cover change is approximately 10 mm year$^{-1}$. In all three subcatchments, the difference is greatest under warmer and wetter scenarios. The difference increases rapidly until 2050 (when the temperature and precipitation modifiers reach their maximum), whereas the increase is slower or partially reversed between 2050 and 2100.

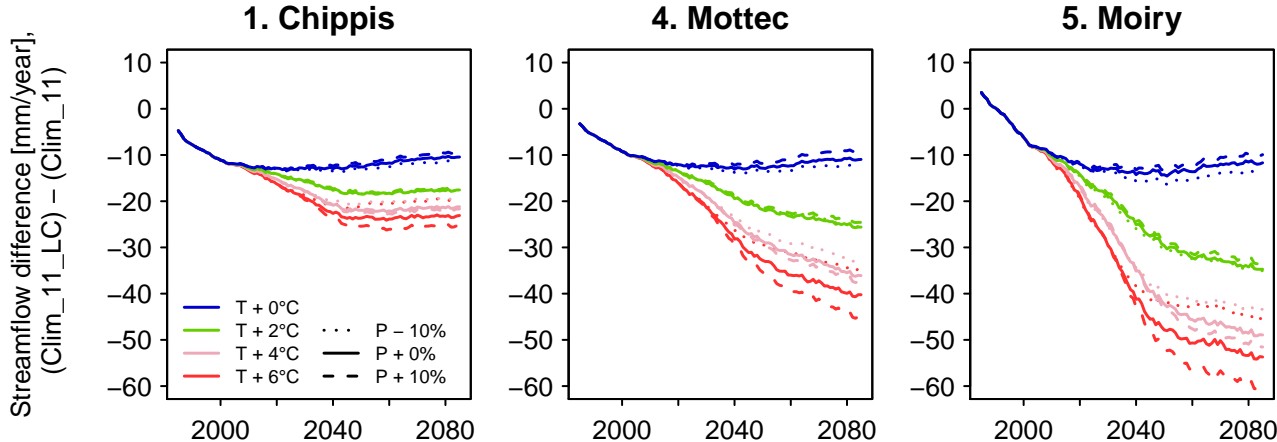

**Figure 11.** Difference in simulated annual streamflow (30-year rolling means) between the standard fully coupled runs (Clim_11) and the coupled runs with land abandonment (Clim_11_LC) for three contrasting subcatchments. In the land abandonment scenario, forest is allowed to grow in all cells classified as "meadows" and "alpine vegetation". The Chippis subcatchment has relatively few meadows, whereas they occupy about a third of the high-elevation catchment Moiry (see Fig. 2). Forest expansion leads to streamflow reduction due to the higher leaf area, and potentially deeper roots. As most meadows are located at high elevations, the effect of forest expansion on streamflow greatly depends on the warming scenario.





## 4 Discussion

### 4.1 Effect of coupling on hydrological simulations

The plausibilization of simulated streamflow (Sect. 3.1) showed that PREVAH in its original version, as well as coupled and uncoupled FORHYCS yielded similar goodness-of-fit scores in the four gauged subcatchments of this study. The differences between standard PREVAH and the two FORHYCS versions are much larger than between coupled and uncoupled FORHYCS. As noted in Sect. 2.1.7, the main difference between standard PREVAH and uncoupled FORHYCS is the parameterization of rooting zone storage capacity SFC. As this difference is considerable (see Fig. S1), the differences in daily streamflow seen on Fig. 5 are to a large extent due to the differing SFC parameterizations. Due to the small differences between the results of coupled and uncoupled FORHYCS, it is not possible to conclude whether varying vegetation properties has led to an improvement of model performance. In the fully coupled version, simulated LAI is quite close to the default values in PREVAH (cf. Fig. 8; standard summer LAI for forests in PREVAH is 8). As seen on Fig. 8, LAI is smaller when simulated with TM_BEK in most elevation bands. A comparison of the simulated streamflow shown in Sect. 3.1 with output from a one-way coupled simulation with TM_BEK (not shown) shows differences of only 3 mm year$^{-1}$ between one-way and two-way couplings, except in the ungauged catchment 1 (Chippis), where the difference is 24 mm year$^{-1}$. This suggests that the sensitivity of simulated streamflow to vegetation properties varies spatially, in line with the findings of Schattan et al. (2013).

The relatively modest effect of the coupling on simulated streamflow, especially in the high-elevation subcatchments, is consistent with the findings of Schattan et al. (2013), whose study domain also included the Navizence catchment. They found a differential effect of transient vegetation parameters on simulated streamflow, with the greatest effects at low elevations (where LAI is much lower than the generic parameter value of the hydrological model) and above the current treeline (where the forest may expand in the future). The issue of scale is also of relevance: as forested cells make up a relatively small fraction of each subcatchment (Fig. 2), even an important change in the water balance of some forested cells will have little influence on catchment-integrated streamflow.

In addition, while leaf area and rooting depth are among the most sensitive vegetation parameters for surface water partitioning (Milly, 1993; Nijzink et al., 2016; Speich et al., 2018a), FORHYCS does not represent all possible impacts of forest dynamics on hydrological processes. For example, forest properties have been related to hydrological model parameters relating to snow (Seibert, 1999) or soil properties (Johst et al., 2008). Badoux et al. (2006) found that forest site type was a good indicator of the dominant runoff processes. While this does not imply a causal relationship between forest characteristics and runoff generation in all cases, some of the differences between runoff processes could be explained by forest properties, such as hydrophobicity of conifer needle litter, which promotes fast runoff processes. On the other hand, forest soils are often associated with low runoff coefficients. Johst et al. (2008) parameterized the soil moisture recharge BETA as a function of land cover type. This parameter, which controls the partitioning of precipitation between plant-available soil moisture and runoff generation, is also used in the local water balance modules of PREVAH and FORHYCS (see Speich et al., 2018a). Speich et al. (2018a) found that BETA was a relatively sensitive parameter for the physiological drought index of FORHYCS. Currently, snow and runoff generation parameters are static in PREVAH and FORHYCS. As they have been obtained through region-





alization, it may be challenging to relate their values to specific forest properties. However, representing the effects of forest dynamics on these processes might further reduce the dependence on calibrated and regionalized parameter values.

## 4.2  Effect of coupling on forest simulations

To assess the effect of various forest-hydrology couplings on the performance of the forest models, several outputs and state
variables were compared against observations. For various reasons, a perfect match between simulated and observed vegetation properties cannot be expected. First, the model simulates the potential natural vegetation dynamics, without considering forest management or disturbances such as fire or avalanches, which again have impacts on stand age. Second, the succession is modeled using bootstrapped meteorological forcing data, which do not contain any trends and may differ greatly from the actual climate in past centuries. Third, the spatial scales of model outputs and observations are not the same. For all these reasons,
the comparison against observations serves as a plausibilization rather than a rigorous validation of the model. Nevertheless, it is assumed that when aggregated to a larger scale, a qualitative comparison with observations can still give some indication of the model's skill. The fully coupled FORHYCS gave reasonable results for biomass, species composition and stand structure (simulated LAI is discussed in the next paragraph). It is important to remember that the coupled and uncoupled forest simulations used two different drought indices, one that depends on transfer variables from PREVAH equations, and one that
is calculated before the simulations, respectively (see Sect. 2.1.3). These indices require two different sets of species-specific drought tolerance parameters. Therefore, it is difficult to assess to what extent the differences in model outputs are due to the coupling, or to the different parameterization. In any case, representing the effect of water availability (and low temperatures) on maximum height greatly improved the simulation of canopy structure, also in uncoupled models. By contrast, reducing leaf area as a function of stress leads to poorer results regarding canopy structure, as well as unrealistic biomass fluctuations.

Simulated leaf area index (LAI) varies greatly between the TreeMig and FORHYCS runs (Fig. 8). The pattern of LAI simulated with TM_BEK across elevation bands follows the distribution of soil moisture storage capacity (Fig. S1). The flat areas at the bottom of the Rhône valley are the only areas where storage capacity exceeds 100 mm. Therefore, LAI is high for the lowest elevation band. By contrast, storage capacity on the slopes is much lower, so that LAI initially sharply decreases with elevation. With TM_RA2015, water is hardly limiting, so that LAI is high at all elevations. LAI simulated with FORHYCS
shows a similar pattern as the remotely sensed data, with an initial increase with elevation, and consistently higher values on north-facing than on south-facing slopes, except at the highest elevations. The absolute values, however, are consistently higher than the observations, sometimes offset by a factor of two. Various factors hinder a direct, quantitative comparison of measured and simulated LAI. First, the 300-by-300 m cells of the remotely sensed dataset may contain non-forested surfaces, such as pastures, clearings, roads or water bodies, even if the cell is classified as forest. The model does not consider this type of spatial
heterogeneity. This is especially relevant in regions with a high spatial variability of land cover, as is the case in this study region. Second, remotely sensed LAI is subject to some uncertainty, due for example to the clumping of needles in coniferous forests (Garrigues et al., 2008). Despite a good overall performance, the authors of the validation report for the Copernicus LAI 300 product (Camacho et al., 2016, p.83) note that forests were under-represented in the validation dataset. Therefore, it is difficult to say to what extent FORHYCS overestimated LAI in this study. Schleppi et al. (2011) measured LAI at 91 forested





sites across Switzerland and used a regression against stand parameters to predict LAI in forests throughout the country. Their measured values range between 1 and 7. They noted a decrease of LAI with elevation, as well as a limitation of LAI due to water availability for sites with annual precipitation below 1000 mm. The values simulated by FORHYCS thus appear plausible at intermediate elevations, where the effects of both water availability and low temperatures are moderate. By contrast, terrestrial

LAI measurements at the bottom of the Rhône valley (Dobbertin et al., 2010) are between 2 and 2.5. Despite the presence of patches with more mesic forest types, especially in the proximity of water bodies (personal observation), such values can be taken as representative for the xeric forests in the Rhône valley. The values simulated by TreeMig and FORHYCS are much higher than this for this elevation band. At high elevations, the decrease in LAI is not as pronounced in the simulations as in the observations (Fig. 8). These results suggest that the spatial variability of LAI is somewhat underestimated by the model,

especially where an environmental factor is particularly limiting.

### 4.3 Effect of coupling on model behavior under climate change

The models used in this study are very similar to those used in the simulation experiment of Lischke and Zierl (2002). In that experiment, they coupled the gap model DisCForM, from which TreeMig was later derived, with a point-scale water balance model which is conceptually similar to the new local water balance module of FORHYCS. They found that the coupling of

forest dynamics and water balance had a stabilizing effect on the simulated system under climate change. In their experiment, coupled simulations converged towards low LAI and lower levels of physiological drought. This effect is not visible to the same extent in the simulations conducted here. The effect of warming on forests is less pronounced in the fully coupled runs, as evidenced by the evolution of streamflow differences on Fig. 10. However, it cannot be excluded that this is due to the different drought tolerance parameters, or to the lesser sensitivity of the FORHYCS drought index to changes in temperature.

In contrast to the study of Lischke and Zierl (2002), FORHYCS includes some additional mechanisms through which the system can react to changes in climate, such as the adaptation of rooting depth and maximum tree height. Some processes may even have a destabilizing influence on the system, such as the high fluctuations in biomass introduced by the stress-induced leaf area reduction.

### 4.4 Effect of additional processes

#### 4.4.1 $CO_2$ concentration

Results of this modeling experiment have shown that an increase in atmospheric $CO_2$ concentration has almost no effect on hydrological processes and vegetation dynamics as modeled by FORHYCS. In this implementation, the physiological effect of elevated $CO_2$ concentration is represented by an additional modifier function to the stomatal resistance parameterization. All else being equal, an increase in $C_a$ leads to an increase in stomatal resistance, and thus to a decrease in potential and

actual transpiration. This slows down canopy water use, and thus leads to lower levels of simulated physiological drought. This formulation does not account for other physiological impacts of elevated $C_a$, such as enhanced photosynthetic rates, or possible acclimation effects. The physiological effect of increased $C_a$ is a source of uncertainty in forest models, due to widely



differing process formulations among models (Medlyn et al., 2011). For example, our results contrast with the simulations of Scherstjanoi et al. (2014), who applied a modified version of LPJ-GUESS and found a crucial influence of $C_a$ on simulated future forest biomass in Switzerland. These differences between models are partly due to the knowledge gaps regarding the underlying processes. According to Medlyn et al. (2011), models that do not consider the physiological effects of $C_a$ at all

are likely to underestimate future forest productivity, whereas some other models are likely to yield overestimates due to an improper representation of other limiting factors. From an ecohydrological point of view, the large-scale effects of increased $C_a$ have been the object of a number of recent studies. For example, Trancoso et al. (2017) found that decreases in streamflow in Australian catchments were caused by vegetation greening, which was in turn driven by elevated $C_a$. These studies suggest that the stomatal effects of increased $C_a$ (transpiration reduction) are more than offset by enhanced vegetation growth. This is

not the case in this study, where the only visible effect was an increase in streamflow, whereas vegetation properties were not affected at all.

### 4.4.2 Land cover change

In this simulation experiment, allowing the forest to grow in areas currently covered by meadows caused a reduction of streamflow of up to 60 mm year$^{-1}$ at subcatchment level (Fig. 11). This is a substantially greater effect than in the simulation

experiment of Schattan et al. (2013), who found a change in annual runoff in the order of 10 mm year$^{-1}$ in regions currently above the treeline as they become forested under simulated climate change. Under the scenario used in their study, temperature was projected to increase by 3-4 K by the end of the century. A major difference with this study is that they only varied LAI, whereas in this study, the development of both LAI and rooting depth were simulated. In the warmest scenarios, LAI and rooting zone storage capacity reached values of 5.5 and 120 mm, respectively, by the end of the simulation even in the

highest elevation band of meadows (cells above 2700 m asl). Both of these variables probably had a major impact on simulated streamflow. These spectacular results must be considered in the light of several potential sources of uncertainty in the model formulation. First, FORHYCS does not account for competition by other vegetation types, which may slow down the expansion of forests. Also, other factors that make the current treelines an extreme environment are not considered by FORHYCS, such as the steep slopes and shallow soils. For example, it was shown that during the warmest period of the Holocene, only

stunted trees were able to establish at high elevations, although the climate would have allowed a forest to grow (Theurillat and Guisan, 2001). Another aspect to consider is that at the beginning of the simulation, meadows and alpine vegetation types were parameterized with a prescribed rooting depth of 22 cm. This value was set arbitrarily, and if it was actually higher for these vegetation types, the hydrological impact of forest expansion would be exaggerated in the simulations.

### 5 Conclusions and Outlook

This study presented a proof-of-concept for a dynamic, spatially distributed model combining hydrological processes and forest dynamics. The main interface variables are leaf area index, rooting depth, as well as a physiological drought index. This model was applied in a case study in a valley with a sharp topographical and hydro-climatic gradient.



The motivation behind developing this model was to apply it to climate change impact studies in which the spatio-temporal forest dynamics and water balance of Switzerland are simulated together. The closer integration of these ecosystem processes would increase the confidence in these model projections, compared to uncoupled models that do not account for changes in the environment besides climate. The research questions were (1) how model coupling impacts the results of simulated

water balance and forest dynamics, (2) which aspects of the coupling were particularly relevant, and (3) how model coupling affects simulation results under climate change. From the hydrological point of view, the coupling had only a modest effect on catchment-integrated streamflow, although this effect was not uniform in space: the greatest effects occurred at low elevations, and in regions currently above the treeline. Regarding forest simulations, model results were compared against multiple data sources to examine model behavior and pinpoint potential weaknesses. In a comparison with a new high-resolution canopy

height dataset, two new indices of agreement between observed and simulated forest structure were developed. This comparison confirmed the importance of specifying an environmental limitation on maximum tree height, as this greatly improved the realism of simulated canopy structure and biomass. Also, a dynamic parameterization of rooting depth led to better model performance. In combination with remotely sensed LAI data, this model-data comparison showed that the coupled model was better able to reproduce observed spatial patterns, although it also highlighted potential deficiencies in the way drought impacts

are represented. Under (idealized) climate change, the forests in the coupled model show greater resilience, which translates into a reduced sensitivity of mean annual streamflow to changes in temperature and precipitation. In some cases, the behavior of the model seems exaggerated, but demonstrates the importance of explicitly modeling relevant processes. This was the case with regard to the possible expansion of forests above the current treeline. On the other hand, the effects of increased $CO_2$ concentration on plant physiology are less than what observations suggest, highlighting the challenges of incorporating

physiological principles into phenomenological models. As these areas are the object of active research, it is expected that new analyses will give the opportunity to test model behavior under these novel conditions, and possibly to improve process formulations.

*Code availability.* Due to the dependency on specific, internally defined data formats, the model cannot be easily transferred to a new environment. For verification purposes, the Fortran code for the processes described in this paper is provided in the supplement.

**Appendix A**

*Author contributions.* HL and MZ designed the study. MJRS and MS wrote the new model code for the integration of PREVAH and TreeMig. MJRS executed the experiments and drafted the manuscript, under supervision from HL and MZ.

*Competing interests.* The authors declare that they have no competing interests





**Table A1.** Values for the species-specific drought tolerance parameter $k_{DT}$ used in this study (last column). The parameter values were obtained by mapping the drought tolerance scores of Niinemets and Valladares (2006) (second column) on the range of values used by Lischke and Zierl (2002). Some of the parameter values were adjusted manually to improve modeled species composition. The parameters of Lischke and Zierl (2002) and of this study indicate the drought index DI at which the drought stress function (Eq. 6) becomes zero. The scores of Niinemets and Valladares (2006) take values between 1 (low tolerance) and 5 (high tolerance) and were obtained based on climatic characteristics at sites where each species was observed. Species marked with an asterisk were excluded from the simulations presented in this paper.

| Species | Drought tolerance parameter in Lischke and Zierl (2002) | Drought tolerance according to Niinemets and Valladares (2006) | Drought tolerance parameter used in this study |
|---|---|---|---|
| Abies alba | 0.37 | 1.81 | 0.28 |
| Larix decidua | 0.45 | 2.31 | 0.42 |
| Picea abies | 0.41 | 1.75 | 0.32 |
| Pinus cembra | 0.43 | 3.01 | 0.45 |
| Pinus montana | 0.40 | 4.23 | 0.49 |
| Pinus sylvestris | 0.55 | 4.43 | 0.45 |
| Taxus baccata | 0.39 | 3.01 | 0.38 |
| Acer campestre | 0.46 | 2.93 | 0.37 |
| Acer platanoides* | 0.42 | 2.73 | 0.36 |
| Acer pseudoplatanus | 0.34 | 2.75 | 0.32 |
| Alnus glutinosa | 0.37 | 2.22 | 0.31 |
| Alnus incana | 0.34 | 1.89 | 0.28 |
| Alnus viridis | 0.37 | 2.48 | 0.33 |
| Betula pendula | 0.35 | 1.85 | 0.28 |
| Carpinus betulus | 0.46 | 2.66 | 0.35 |
| Castanea sativa* | 0.30 | 3.46 | 0.42 |
| Corylus avellana | 0.46 | 3.04 | 0.38 |
| Fagus sylvatica* | 0.37 | 2.40 | 0.33 |
| Fraxinus excelsior | 0.39 | 2.50 | 0.34 |
| Populus nigra | 0.30 | 2.20 | 0.31 |
| Populus tremula | 0.41 | 2.85 | 0.37 |
| Quercus petraea | 0.44 | 3.02 | 0.38 |
| Quercus pubescens | 0.40 | 4.10 | 0.48 |
| Quercus robur | 0.44 | 2.95 | 0.38 |
| Salix alba | 0.39 | 2.00 | 0.29 |
| Sorbus aria | 0.38 | 3.55 | 0.43 |
| Sorbus aucuparia | 0.27 | 2.11 | 0.30 |
| Tilia cordata | 0.40 | 2.75 | 0.36 |
| Tilia platyphyllos | 0.40 | 2.52 | 0.34 |
| Ulmus scabra | 0.35 | 2.41 | 0.33 |





*Acknowledgements.* This research was funded by the Swiss National Science Foundation (no. 153544) and the Swiss Federal Office for the Environment (no. 15.0003.PJ/Q104-0149). The authors would like to thank James Kirchner (ETH Zurich) and Giorgio Vacchiano (University of Milan) for helpful comments on a previous version of this paper. Furthermore, we would like to thank Esther Thürig for providing the forest inventory data, and Christian Ginzler for providing the canopy height data.



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
