# Peer review of "FORHYCS v1.0: A spatially distributed model combining hydrology and forest dynamics"

_Geoscientific Model Development, 2019_

## Referee Comment (RC1) · Anonymous Referee #1 · 26 Jul 2019

**Review of "FORHYCS v1.0:** A spatially distributed model combining hydrology and forest dynamics" by Speich et al.**

The study of Speich et al. couples the hydrological model PREVAH with the forest model TreeMig, which is together presented as the new model FORHYCS. The study area is the Navizence catchment in Switzerland and several modelling scenarios are applied here. The modelling scenarios consist of one group of scenarios that models the full succession, and one that models under climate change. Additionally, the model is run in uncoupled and coupled mode with different settings switched on and off. The authors conclude that including drought-induced height limitations for vegetation improves model performance, but the effects on streamflow by coupling the models was limited. Coupling the model also led to reduced sensitivity of streamflow to increased temperatures and precipitation in the climate change scenarios.

The authors present a thorough analysis in a generally well-written manuscript. The work seems solid, and all the different model scenarios seemed a large effort to me. However, I do have several comments that the authors may need to improve on.

**Major Comments**

My first comment is more a question out of curiosity, as the overestimation of LAI by the model intrigues me. It will strongly depend on the formulation of leaf area dynamics in Eq. 1, so how confident are you that this equation (or more the parameterization of this equation) is correct? How many trees were for example used to derive the allometric relations? Besides, the species specific parameters (like SLA, and a1,a2) are not reported, also not in the Supplement, so can you add these? So in general, could your leaf area formulation be the reason for the observed over-estimation?

I am also a bit confused by equation 3. The fractional cover in LSM's or remote sensing products is often related related to LAI by the Lambert-Beer relation: FC = 1 - exp(-K \* LAI), where LAI is the total leaf area index (or crown index), FC is fractional cover, K is an extinction coefficient, (e.g. Bréda, 2003; Choudhury, 1987; Monsi, 2004). The extinction coefficient is a function of leaf inclination and often set to 0.5. Here, this seems to be set to 1 for all species, which seems a bit high, is that correct? In addition, why are the exponents summed? Shouldn't you just add up the different final fractional covers of the species when the area stays the same? This is also what you describe on page 17 (if I am not mistaken), where you take the cumulative sums of the classes.

In addition, LAI and fractional cover are compared by two newly developed error measures, which only compare on one specific moment in time. However, getting the seasonality right in these models is quite important, and one of the minimum things the model should be able to represent is the seasonal signal. Did you compare the timeseries of simulated and observed LAI? It's rather simple to do, and, in my view, provides much more information then the error measures as presented by the authors. So how well is the seasonality captured by the model?

I am also a bit confused on how the effect of elevated  $CO_2$  is studied. How can you evaluate the effect of elevated  $CO_2$  if you switch off the stomatal response to high  $CO_2$  (P22.L3)? Do you mean you keep the conductance the same? It would be much more interesting to keep the feedbacks in place, so why you do this? However, later on in the manuscript, the stomatal conductance is discussed, so can you clarify what you do exactly?

I am not too familiar with PREVAH, unfortunately, but the authors state that the model structure is similar to HBV. That would mean there are also several parameters that do not relate to vegetation (such as recession parameters, routing, snow parameters), so how are these determined? It can also be seen in Figure 5 that snow melt and recessions are quite off compared to the observations, which is probably just due to the remaining parameters. It may also affect the conclusions based on the climate change scenarios, as the snow melt is highly affected by the temperature changes.

My last, but most important comment is on several key-findings which do not seem to be (entirely) supported by data. For example, one of the key findings presented in the manuscript concerns the effect of the climate change scenarios on streamflow. However, the result of only one specific catchment is shown in Figure 10, how do the results for the other catchments look like? Similarly, an analysis on elevated CO2-levels is described, but no results are shown in any of the graphs. Please add some graphs and evidence to support the statements you make here.

Generally, I enjoyed reading the manuscript, but some revisions may be necessary. I hope the authors find my comments useful, and I look forward to an improved version of the manuscript.

**Minor comments**

I would like to suggest to change names of the modelling scenarios into more meaningful names, or add clarifications in the text when discussing a certain scenario. Names like Succ\_TM\_BEK, or T6\_P-10, are not very informative and make it hard to understand what happens without looking at the table all the time.

P8.L8-9. In this way, the equation does not seem consistent in units. What is the unit of Pd,sp?

P8.L10. This seems a rather arbitrary number to me, why 833 m2?

P8.L18-20. How does crown area relate to leaf area?

P9.L25-30. So the used transpiration values are model outputs, correct?

P10.L9. Is fDS not a single yearly value, as DI is a single year value too? What do you use to calculate the geometric mean in that case?

P10.L18. "i.e. with a decrease...is at kDT", I am not sure I follow, can you please clarify?

P11.L1. he -> the

P11.L26. Why did you use these numbers? Seems a bit arbitrary.

P13. So  $PP_0$  includes the carbon costs? What are these values based on?

P16.L1-5. If a large amount is diverted by pipelines, can you compare modelled and observed discharge? Which sub-catchments are affected by this?

P16.L27. "As the sampling plots... a larger area." This sentence is a bit unclear to me, what do you mean?

P17.L19. Please correct reference.

P22.L31 The plot only shows Acer spp., so how can I see this?

P24.L21. This sounds a bit counter-intuitive, wouldn't you expect a higher biomass when there is no LAI-reduction? What is the reason for this?

P29.L10-11. Is this not counter-intuitive too? You would expect (also because of equations 7 and 8, where additional carbon is allocated for roots under stress) that the roots will go deeper in case of drier scenarios, and that LAI would go down, correct?

P31.L4-5. You described earlier that Eq. 12 was set to 1, correct?

P32.L4-14. How different are the landuses eventually at the end of the runs? Are the differences mainly due to different forest covers under the different scenarios?

P33.L14. Based on the data as shown, you cannot claim that the sensitivity of streamflow to vegetation properties varies spatially. This would also mean you need to have the same vegetation in different places and observed different changes in streamflow, but I believe that is not the case.

P37.L7-8. "the greatest effects occurred at low elevations, and in regions currently above the treeline", where do you show this? Please back this up with some evidence in the main manuscript, especially when it is a key-finding.

Eq3. Please define all variables and subscripts

Table1. There two Succ\_noHmax-scenarios, please correct.

Table 2. Please describe the abbreviations in the caption or replace them with a description.

Fig1a. Please define SFC also in the figure.

Fig5. There are a couple of things that seem a bit odd to me in this plot. In Fig 5a, Prevah seems to be much closer to the observations then the other two model set-ups, but in Table 2 the KGE-values are lower. Is that correct? In addition, Forhycs00 and Forhycs11 are on top of each other in Figure 5a, whereas Figure 5b suggests a difference of up to 0.3 m3/s.

Code availability: I would suggest to share your code on github or gitlab, instead of the supplement. Please also add links to the actual datasets used in the study.

Appendix A: Why is there an appendix in the main manuscript and also a supplement? Should Table A1 not just be part of the Supplement then?

**References**

Bréda, N.J.J., 2003. Ground based measurements of leaf area index: a review of methods, instruments and current controversies. J Exp Bot 54, 2403–2417. https://doi.org/10.1093/jxb/erg263

- Choudhury, B.J., 1987. Relationships between vegetation indices, radiation absorption, and net photosynthesis evaluated by a sensitivity analysis. Remote Sensing of Environment 22, 209– 233. https://doi.org/10.1016/0034-4257(87)90059-9
- Monsi, M., 2004. On the Factor Light in Plant Communities and its Importance for Matter Production. Annals of Botany 95, 549–567. https://doi.org/10.1093/aob/mci052

---

## Referee Comment (RC2) · Anonymous Referee #2 · 7 Aug 2019

The article presents how a modified version of hydrological model, PREVAH, is coupled with a forest landscape model, TreeMig, that makes the distributed ecohydrological model, FORHYCS. The FORHYCS is applied in a mountainous catchment to illustrate the coupling effect on hydrology and forest structure along with a series of climate change experiment.

The authors did a great job collaborating on all the information from various sources and made up a good modeling concept. My main concern about the research is, this study area is not an ideal watershed for doing the experiments. Since we are talking about coupling a forest model with a hydrological model, we better have a catchment

that is covered with forest. The five subcatchments of Navizence, have ~30% forested area in Chippis (but no streamflow data) and ~15% forested area in Vissoie. The other three catchments have negligible fraction of forested area. That might be part of the reason that we don't see much hydrograph difference between uncoupled and coupled runs even in the most forested catchment Vissoie. In fact the uncoupled-coupled streamflow differences are smaller in the other less forested catchments.

It would be great if authors could acquire downscaled meteorological data from latest climate modeling scenarios and test the FORHYC with those time series data. It's not that the delta method (e.g. ±degrees and/or precipitation) isn't scientifically sound, but the climate modeling data provides more variation and insight to the change of climate.

Specific comments P1L12-13: give names of the two new metrics.

P2L22: what aspects of mountainous regions are sensitive to what type of global change?

P8L7: explain AI, is it the stand leaf area?

P9L3: define EDI, is it Evaporative Demand Index?

P10L5: again, define fDS.

P14L7: forest minimum stomatal resistance at 180 s m-1 seems to be at the lower end of what has been reported. The stomatal/canopy resistance could be one of the most influential parameters when it comes to estimating transpiration (not sure about this PREVAH model). Why is a single number resistance superior to the previous "minimum canopy resistance for each land cover class"? In any case, the number needs to be justified according to the region and species being applied in this particular study.

P22L9-23: I've got confused by the streamflow simulations. How was the model calibrated to generate these KGE scores? I assume the PREVAH/FORHYCS were "tuned" to their best performance before the series of experiment. It seems three modeling runs generated model efficiencies that vary from catchment to catchment. Would some

other combinations of parameterization make the results look differently?

P23-Figure 5: I'd suggest giving PREVAH a solid light grey line to make it easier to read the difference between FORHYCS00 and FORHYCS11 (which matters more).

P26L8-20: labels from different simulations need to be unified. For example, Succ_TM_BEK is BEK in the Figure 8?

P33L3-15: looks like a large part of the reason that uncoupled and coupled modeling runs were making rather subtle differences is, the catchment areas are not forested enough for the forest model to pass signals back to the hydrology module. If we look at the hydrologic modeling performance for catchments, Vissoie has more vegetated areas (other than Chippis, which has no observations), thus has the lowest KGE score. The other three catchments, none with meaningful forest fraction, perform much better without interplay with the forests.

—————————————————

---

## Author Comment (AC2) · 12 Aug 2019

We would like to thank the reviewer for their thorough response to our manuscript. Their comments will be very helpful to improve our manuscript. We are glad to take the opportunity of this discussion format to address the points they raise.

**My main concern about the research is, this study area is not an ideal watershed for doing the experiments. Since we are talking about coupling a forest model with a hydrological model, we better have a catchment that is covered with forest. The five subcatchments of Navizence, have 30% forested area in Chippis (but no streamflow data) and ~15% forested area in Vissoie. The other three catchments have negligible fraction of forested area. That might be part of the reason that we don't see much hydrograph difference between uncoupled and coupled runs even in the most forested catchment Vissoie. In fact the uncoupled-coupled streamflow differences are smaller in the other less forested catchments**

The relative fractions of currently forested, potentially forested and never-forested area are shown on Fig. 2. The fractions of forested area are higher than the numbers stated in the review: more than half of the Chippis subcatchment is currently forested (> 25 km$^2$ out of ~50 km$^2$), and for Vissoie, this number is about 25% (~15 out of ~60 km$^2$).

The abandonment of pastures and replacement with forests is an ongoing process in this region, and is expected to continue given current socio-economic trends (see Price et al. 2016). Therefore, the potentially forested area is also relevant, and the behavior of the model for this situation is also tested in this study. For the five subcatchments of this study, the fractions of currently and potentially forested area are ~ 90% (Chippis), 66% (Torrent du Moulin), 70% (Vissoie), 25% (Mottec), 33% (Moiry/Lona).

Considering also the other reasons for selecting this study region, listed in Section 2.2 (p.14f) (strong ecological and hydro-climatic gradient due to elevation differences), we argue that this is a suitable study region for testing the model. Another study is planned in which we apply FORHYCS to several catchments representing different hydro-climatic regions of Switzerland.

**It would be great if authors could acquire downscaled meteorological data from latest climate modeling scenarios and test the FORHYC with those time series data. It's not that the delta method (e.g. ±degrees and/or precipitation) isn't scientifically sound, but the climate modeling data provides more variation and insight to the change of climate.**

In the framework of an ongoing study, we have acquired the CH2018 scenarios (NCCS 2018), based on the EURO-CORDEX simulations. Given the scope of this study, it does not seem necessary to run FORHYCS with all 39 model chains. Instead, we suggest providing results for the three chains selected by Brunner et al. (2019) as representative for dry, medium and wet conditions (in addition to the delta runs).

**P1L12-13: give names of the two new metrics.**

The names for these metrics (dH95 and 1-ABC) are introduced in Section 2.2.3.

**P2L22: what aspects of mountainous regions are sensitive to what type of global change?**

We will be more specific on this in a revised version and mention the following aspects:

- Strong influence of snow storage on hydrology, which may change with warming.
- Warmer and drier conditions may alter species composition in temperature-limited ecosystems.
- Upwards shift of treeline.

**P8L7: explain Al, is it the stand leaf area?**

Yes – we will include this in the text.

**P9L3: define EDI, is it Evaporative Demand Index?**

This is a remainder of an old version of the text, thank you for catching this. This refers to the drought index used in TreeMig, and described in the rest of the paragraph. The symbol "EDI" will be removed.

**P10L5: again, define fDS.**

Vitality reduction function due to drought stress. Based on the comments of Reviewer #1, this passage will need to be clarified. We will also consider this remark.

**P14L7: forest minimum stomatal resistance at 180 s m-1 seems to be at the lower end of what has been reported. The stomatal/canopy resistance could be one of the most influential parameters when it comes to estimating transpiration (not sure about this PREVAH model). Why is a single number resistance superior to the previous "minimum canopy resistance for each land cover class"? In any case, the number needs to be justified according to the region and species being applied in this particular study.**

This concerns only uncoupled FORHYCS, i.e. a simultaneous run of the hydrological and forest parts of the model without any transfer of information between them. For coupled FORHYCS, minimum stomatal resistance is parameterized according to species and tree height, as described in Section 1.3.2 of the Supplement (which we forgot to reference from the main text – this will be fixed in a new version).

Uncoupled FORHYCS is not meant to represent an improvement from stand-alone PREVAH. Rather, its purpose here is to provide a comparable baseline against which to assess the effect of the coupling. This was not done directly against stand-alone PREVAH for the reason stated in Section 2.1.7 (PREVAH uses a different dataset for soil water storage capacity which is incompatible with the new water balance module).

In another study (Speich et al. 2018a), we found that RSMIN was a rather influential vegetation property for simulated annual water balance (though not as influential as soil water storage capacity or LAI). This was one of the motivations for coupling the models. For uncoupled FORHYCS, the value of 180 s/m was selected as a standard value from the literature (see the cited reference of Guan and Wilson 2009, who base their choice on the review by Körner (1994)). For comparison, in another study in a region nearby

(Peters et al. 2019), stomatal resistances of 128 s/m and 285 s/m were found for *Larix decidua* and *Picea abies*, respectively.

**P22L9-23: I've got confused by the streamflow simulations. How was the model calibrated to generate these KGE scores? I assume the PREVAH/FORHYCS were "tuned" to their best performance before the series of experiment. It seems three modeling runs generated model efficiencies that vary from catchment to catchment. Would some other combinations of parameterization make the results look differently?**

As Reviewer #1 had a similar comment, here is our answer to their question regarding calibration:

Indeed, there are various parameters related to non-vegetation aspects. Some of these parameters are constant for the whole study area; others are spatially variable and have a different value for each grid cell. The spatially variable parameter values were determined in a previous study (Zappa and Bernhard 2012) based on the regionalization method of Viviroli et al. (2009). The spatially constant parameters were also taken from previous studies. As a different dataset for soil water holding capacity was used in this study (see Section 2.2.2), some parameters were manually adjusted to improve the optical fit of the streamflow lines. Due to the proof-of-concept nature of this study, a full calibration was not undertaken.
In a revised version, we will provide the reference for the spatially variable parameter values, as well as the values used for the spatially constant parameters, in the Supplement.

**P23-Figure 5: I'd suggest giving PREVAH a solid light grey line to make it easier to read the difference between FORHYCS00 and FORHYCS11 (which matters more).**

OK

**P26L8-20: labels from different simulations need to be unified. For example, Succ_TM_BEK is BEK in the Figure 8?**

Based on comments from Reviewer #1, we will reconsider the naming of the different runs. We will also take this comment into account.

**P33L3-15: looks like a large part of the reason that uncoupled and coupled modeling runs were making rather subtle differences is, the catchment areas are not forested enough for the forest model to pass signals back to the hydrology module. If we look at the hydrologic modeling performance for catchments, Vissoie has more vegetated areas (other than Chippis, which has no observations), thus has the lowest KGE score. The other three catchments, none with meaningful forest fraction, perform much better without interplay with the forests.**

It is plausible that the degree of forest cover had some influence for the KGE scores. However, I would also see it in the view of the different soil parameterizations used in PREVAH and uncoupled FORHYCS. Those differ in currently and potentially forested areas, so that lower performance is to be expected in those areas.

It is also worth considering that streamflow from the Mottec and Moiry subcatchments stems to a large extent from the glaciers, the water balance of which the model is able to simulate relatively well.

**References**

Brunner, MI,  Björnsen Gurung, A, Zappa M,  Zekollari, H, Farinotti D, Stähli, M (2019): Present and future water scarcity in Switzerland: Potential for alleviation through reservoirs and lakes. Science of The Total Environment 666, 2019, 1033-1047, https://doi.org/10.1016/j.scitotenv.2019.02.169

Guan, H. and Wilson, J. L. (2009): A hybrid dual-source model for potential evaporation and transpiration partitioning, Journal of Hydrology, 377, 5 405–416, https://doi.org/10.1016/j.jhydrol.2009.08.037

Körner, C (1994): Leaf diffusive conductances in the major vegetation types of the globe. E.D. Schulze, M.M. Caldwell (Eds.), Ecophysiology of Photosynthesis, Ecological Studies v.100, Springer-Verlag (1994), pp. 463-490

National Centre for Climate Services (NCCS) (2018): CH2018 — Climate Scenarios for Switzerland Technical Report, NCCS, Zurich (2018)

Peters, RL, Speich, M, Pappas, C, et al. (2019) Contrasting stomatal sensitivity to temperature and soil drought in mature alpine conifers. *Plant Cell Environ*. 2019; 42: 1674– 1689. https://doi.org/10.1111/pce.13500

Price, B., Kaim, D., Szwagrzyk, M., Ostapowicz, K., Kolecka, N., Schmatz, D. R., Wypych, A., and Kozak, J. (2016): Legacies, socio-economic and biophysical processes and drivers: the case of future forest cover expansion in the Polish Carpathians and Swiss Alps, Regional Environmental Change, https://doi.org/10.1007/s10113-016-1079-z

Speich, M. J., Zappa, M., and Lischke, H. (2018a): Sensitivity of forest water balance and physiological drought predictions to soil and vegetation parameters - A model-based study, Environmental Modelling & Software, 102, 213–232, https://doi.org/10.1016/j.envsoft.2018.01.016

Viviroli, D, Mittelbach, H, Gurtz, J, Weingartner, R (2009): Continuous simulation for flood estimation in ungauged mesoscale catchments of Switzerland – Part II: Parameter regionalisation and flood estimation results. Journal of Hydrology 377, 1-2, 208-225, https://doi.org/10.1016/j.jhydrol.2009.08.022

Zappa, M, and Bernhard, L (2012): Klimaänderung und natürlicher Wasserhaushalt der Grosseinzugsgebiete der Schweiz. Technical report. Birmensdorf, Eidg. Forschungsanstalt WSL.

---

## Short Comment (SC1) · 14 Aug 2019

This comment addresses the compliance of this manuscript with the GMD policy on code and data availability. The issues raised here must be satisfactorily addressed before a revised manuscript can be accepted for publication.

This paper presents a new coupled model. The source code of this model is substantially absent, with seemingly only a few files that have been particularly modified being presented in the supplement. The excuse provided is that the model would be of limited use to others due to internally defined data formats. This excuse does not address the reasons that source code is required for GMD publication. These are addressed

in detail in the recent GMD editorial[1], however the core point is that it is only with access to the source code that a reader can actually determine what a model really does. Providing a few exemplary fragments does not satisfy this.

On the specific subject of file formats, the manuscript types page of the GMD website[2] is explicit that manuals should also be provided. At a minimum one would expect that any input or output file formats would be adequately documented.

The full model source code, as well as any other files required to run the examples presented in the manuscript, needs to be placed in a persistent public archive in line with GMD policy, and appropriately cited from the code availability section.

[1]ttps://doi.org/10.5194/gmd-12-2215-2019

[2]https://www.geoscientific-model-development.net/about/manuscript_types.html

---

## Short Comment (SC2) · 26 Aug 2019

We are in the process of uploading the complete FORHYCS code (including a manual) to our institution's repository (https://www.envidat.ch). The code will be assigned a DOI published under an open license. However, we will need some additional time to finalize this process (about one or two weeks). We hope that this will not disturb the editorial process too much.

Best regards

Matthias Speich

---

## Author Response (AR1)

Editorial board
Geoscientific Model Development

Matthias Speich
Swiss Federal Research Institute WSL
Zürcherstrasse 111
CH-8903 Birmensdorf

Birmensdorf, November 19, 2019

Concerns: Model Description Article for submission to Geoscientific Model Development

Dear Editor,

It is our pleasure to submit a revised version of our manuscript:

**FORHYCS v1.0: A spatially distributed model combining hydrology and forest dynamics**

by Matthias Speich, Massimiliano Zappa, Marc Scherstjanoi and Heike Lischke

We are grateful for the two thorough reviews of our manuscript and believe that we were able to address all of the reviewers' concerns. The manuscript now also includes model runs with meteorological forcing from the climate models of the EURO-CORDEX experiment. Also, to comply with GMD's guidelines, the model code is now publicly accessible on a permanent repository. Finally, the reviewers have pointed out several incomplete and unclear formulations, which we haven now corrected and clarified.

Please find, in this document, out point-by-point response to the two reviewers and a marked version of our manuscript, tracking all modifications from the previous version.

We are looking forward to reading your response.

Kind regards from Switzerland, on behalf of all authors

Matthias Speich
matthias.speich@wsl.ch

*This is an update to the response submitted to the comments of Reviewer #1 in the interactive discussion. This version also describes how the changes were included in the revised manuscript. All changes to the original response document are marked in italic. Page and line numbers of the new manuscript refer to the marked version.*

We would like to thank the reviewer for their thorough response to our manuscript. Their comments will be very helpful to improve our manuscript. We are glad to take the opportunity of this discussion format to address the points they raise.

**My first comment is more a question out of curiosity, as the overestimation of LAI by the model intrigues me. It will strongly depend on the formulation of leaf area dynamics in Eq. 1, so how confident are you that this equation (or more the parameterization of this equation) is correct? How many trees were for example used to derive the allometric relations? Besides, the species specific parameters (like SLA, and a1,a2 ) are not reported, also not in the Supplement, so can you add these? So in general, could your leaf area formulation be the reason for the observed over-estimation?**

The equation for leaf area and its parameterization are taken directly from the original TreeMig. They have originally been parameterized by Bugmann (1994, 1996) for the gap model FORCLIM, which shares many process formulations with TreeMig. The basis for the parameterization is the dataset collected by Burger (1945 - 1953), consisting of measurements of tree height, diameter and leaf area on 583 trees of five species or species groups. Tree species not represented in the dataset are assigned to one of the represented species, still following Bugmann (1994). The number of trees for each species, and specific parameters are given in the appendix of Bugmann (1994). Since this document is not widely available, we will repeat this information in a new version of the Supplement.

*Section 1.3 of the Supplement now contains the parameter values for the allometric leaf area function, as well as some background information on the data source.*

**I am also a bit confused by equation 3. The fractional cover in LSM's or remote sensing products is often related related to LAI by the Lambert-Beer relation: FC = 1 – exp( -K * LAI), where LAI is the total leaf area index (or crown index), FC is fractional cover, K is an extinction coefficient, (e.g. Bréda, 2003; Choudhury, 1987; Monsi, 2004). The extinction coefficient is a function of leaf inclination and often set to 0.5. Here, this seems to be set to 1 for all species, which seems a bit high, is that correct? In addition, why are the exponents summed? Shouldn't you just add up the different final fractional covers of the species when the area stays the same? This is also what you describe on page 17 (if I am not mistaken), where you take the cumulative sums of the classes.**

The calculation of fractional cover in FORHYCS is independent from the calculation of LAI. Instead, it is based on crown area, which is calculated from tree height using species-specific empirical relationships (the formula and its species-specific coefficients are reported in the supplementary material of Zurbriggen et al. (2014), Section B5). This way, it is not necessary to estimate an extinction coefficient.

The procedure used here and in Zurbriggen et al. (2014) was originally developed by Crookston and Stage (1999). It is based on the assumption that trees are randomly distributed in space (which is consistent with the way light penetration is calculated in TreeMig) and accounts for overlap between crowns. On page 17, the same procedure is applied. For example, applying Eq. 3 to the upper 3 height classes will return the fractional cover for the trees belonging to these classes, accounting for overlap between them. This assumes that shading of lower parts of the crowns by smaller trees can be neglected.

In a revised version of the manuscript, these two assumptions (random distribution and no shading by shorter trees) will be explicitly stated in Section 2.2.3 (currently p. 17). In addition, to clarify that the same procedure is being used, a modified version of Eq. 3 will be introduced in the same section:

$$f_{c,i} = \left[1 - exp\left(-1 \times \sum_{sp=1}^{nspc} \sum_{hc=i}^{nhcl} (n_{sp,hc}/833) \times CA_{sp,hc}\right)\right],$$

where $f_{c,i}$ is the fractional cover of the $i$ upper height classes.

*In the new manuscript, it is pointed out explicitly that LAI and fractional cover are calculated independently from each other (p8 l23). Also, the calculation of cumulative fractional cover by height class and the assumptions behind it are explicitly described, as discussed above (p18 l23).*

**In addition, LAI and fractional cover are compared by two newly developed error measures, which only compare on one specific moment in time. However, getting the seasonality right in these models is quite important, and one of the minimum things the model should be able to represent is the seasonal signal. Did you compare the timeseries of simulated and observed LAI? It's rather simple to do, and, in my view, provides much more information then the error measures as presented by the authors. So how well is the seasonality captured by the model?**

Indeed, these metrics focus on the canopy structure at full foliage cover. This is intentional, as their purpose is to evaluate the forest structure and improvements over stand-alone TreeMig (hence the comparison with TreeMig in Fig. 7 and 8).

An evaluation of the intra-annual variations in leaf area was not carried out for the following reasons:

- As seen on Fig. 6, the observed and simulated distribution of species do not match well (for the reasons discussed in Section 4.2, p. 34 l 4-9). Therefore, a good fit to phenological observations is not to be expected.
- In another study (Speich et al., 2018a), the sensitivity of a water-balance model (corresponding to the surface water balance part of FORHYCS) to vegetation properties was assessed. It was found that long-term water partitioning was not very sensitive to growing season length (defined as the number of days with full foliage), as compared to LAI at full foliage. As long-term annual streamflow is the main hydrological output of interest in this case study, we chose not to evaluate simulated phenology in detail.
- While they do not constitute a validation, the error metrics obtained during the calibration of the phenology submodel (reported in Tables S3 and S4 in the Supplement) give an indication of its strength. As discussed in the Supplement, spring phenology can be reproduced reasonably well in most cases, whereas

autumn phenology is more problematic. This is consistent with other studies where empirical phenological models were applied.

**I am also a bit confused on how the effect of elevated CO2 is studied. How can you evaluate the effect of elevated CO2 if you switch off the stomatal response to high CO2 (P22.L3)? Do you mean you keep the conductance the same? It would be much more interesting to keep the feedbacks in place, so why you do this? However, later on in the manuscript, the stomatal conductance is discussed, so can you clarify what you do exactly?**

Our formulation here may indeed be unclear. In most simulation runs, the effect of elevated $CO_2$ on stomatal resistance (Eq. 12) is active, i.e. stomatal resistance is impacted by atmospheric $CO_2$ concentration. As this is a new addition (there is no $CO_2$ effect either in PREVAH or in the water balance model described in Speich et al. (2018a)), the purpose of the NCS runs is to test the strength of this effect. Therefore, in the NCS runs, the stomatal effect of $CO_2$ is switched *off*, i.e. Eq. 12 is set to 1. Comparing the NCS runs with the standard runs will give an indication of how strong the $CO_2$ effect in the model is.

*This is now clarified (p21 l20).*

**I am not too familiar with PREVAH, unfortunately, but the authors state that the model structure is similar to HBV. That would mean there are also several parameters that do not relate to vegetation (such as recession parameters, routing, snow parameters), so how are these determined? It can also be seen in Figure 5 that snow melt and recessions are quite off compared to the observations, which is probably just due to the remaining parameters. It may also affect the conclusions based on the climate change scenarios, as the snow melt is highly affected by the temperature changes.**

Indeed, there are various parameters related to non-vegetation aspects. Some of these parameters are constant for the whole study area; others are spatially variable and have a different value for each grid cell. The spatially variable parameter values were determined in a previous study (Zappa and Bernhard 2012) based on the regionalization method of Viviroli et al. (2009). The spatially constant parameters were also taken from previous studies. As a different dataset for soil water holding capacity was used in this study (see Section 2.2.2), some parameters were manually adjusted to improve the optical fit of the streamflow lines. Due to the proof-of-concept nature of this study, a full calibration was not undertaken.
In a revised version, we will provide the reference for the spatially variable parameter values, as well as the values used for the spatially constant parameters, in the Supplement.

*The source for the parameter values used is now given in Section 1.7.2 of the Supplement (referred to from p.17 l.5 in the main text).*

**My last, but most important comment is on several key-findings which do not seem to be (entirely) supported by data. For example, one of the key findings presented in the manuscript concerns the effect of the climate change scenarios on streamflow. However, the result of only one specific catchment is shown in**

**Figure 10, how do the results for the other catchments look like? Similarly, an analysis on elevated CO2-levels is described, but no results are shown in any of the graphs. Please add some graphs and evidence to support the statements you make here.**

We agree that these figures are necessary to give the full picture. At the end of this document, we include the future streamflow projections and differences between model configurations (equivalent to Fig. 10) for the other four catchments, as well as the differences in streamflow for model runs with and without $CO_2$ effect on stomatal resistance. In a revised version, we will include these figures in the Supplement and refer to them in the Discussion.

*The Supplement now contains figures for mean annual streamflow under future scenarios for subcatchments 2-5 (S18-S21), and for a comparison of streamflow with and without the effect of CO2 concentration (S22).*

**Minor comments**

**I would like to suggest to change names of the modelling scenarios into more meaningful names, or add clarifications in the text when discussing a certain scenario. Names like Succ_TM_BEK, or T6_P10, are not very informative and make it hard to understand what happens without looking at the table all the time.**

We welcome this comment, as this may indeed be a factor that makes it difficult to follow the text. We will take this into consideration when submitting a new version, and modify the text and figures accordingly.

*The names of runs have been revised (Table 1) and the use of run names is now consistent throughout the manuscript. Also, a new paragraph explains the nomenclature for the different model runs (p.19 l.19).*

**P8.L8-9. In this way, the equation does not seem consistent in units. What is the unit of Pd,sp?**

The phenological status $p_{d,sp}$ is dimensionless and ranges from 0 to 1. Its purpose is to scale leaf area when the foliage is not fully developed (in autumn, winter and spring). We forgot to specify the unit [-] for this variable in the text and will correct this in a revised version.

*This has been clarified (p.8 l.10).*

**P8.L10. This seems a rather arbitrary number to me, why 833 m2 ?**

This number has its origin in the gap model FORCLIM (Bugmann 1994, 1996), from which many process formulations of TreeMig were taken. In FORCLIM, 833 $m^2$ is the reference area of a simulated forest plot (roughly equivalent to the crown area of a large, dominant tree).

*This has been clarified (p.8 l.12).*

**P8.L18-20. How does crown area relate to leaf area?**

As noted in our response to the second comment, leaf area and crown area are calculated independently from each other, both using empirical relationships with tree size.

**P9.L25-30. So the used transpiration values are model outputs, correct?**

Yes, both actual and potential transpiration are simulated in the surface water balance part of the model.

*This has been clarified (p.9 l.29).*

**P10.L9. Is fDS not a single yearly value, as DI is a single year value too? What do you use to calculate the geometric mean in that case?**

Indeed, there is some information missing here to properly follow. Modeled tree growth depends on an environment-dependent function ranging from 0 (maximum stress) to 1 (unstressed conditions). This function is the geometric mean of three functions:

- The drought stress function fDS (Eq. 6)
- The effect of degree-day sum (Eq. S8 in the supplement)
- A stress function describing the effect of nitrogen supply

The last function is not mentioned in the manuscript, as the nitrogen supply is kept constant over the whole study area and period. Nevertheless, as this part of the model cannot be described without this function, it will be included into the supplement.

*The description of the stress functions has been corrected and clarified in the main text (p.10 l.11 and following) and in the Supplement (Section 1.2).*

**P10.L18. "i.e. with a decrease...is at kDT", I am not sure I follow, can you please clarify?**

There is a mistake in this sentence – this is not about LAI, but tree height. The correct version is (also modified for additional clarity): "The former is parameterized following Rasche et al. (2012), i.e. species-specific maximum tree height may be reduced as a function of the bioclimatic indices DI and DDEGS. The parameter $k_{redmax}$, which is also species-specific, indicates the fraction of maximum height that can be attained by trees if one of the environmental vitality functions is at its minimum. The more severe of the two reductions (drought or degree-days) is applied."

*The text has been clarified as described above (p.10 l.23 and following).*

**P11.L1. he –> the**

Thank you – this will be corrected.

**P11.L26. Why did you use these numbers? Seems a bit arbitrary.**

First, we noticed that there was a mistake in the way Eq. 12 is reported. The parameter $j_c$ is equivalent to (1-a) in Medlyn et al. (2001) (their Eq. 5). Therefore, the correct version of Eq. 12 is:

$$f_5 = \left(1 - j_c \left(\frac{\min(C_a, 700)}{350}\right) - 1\right)^{-1}.$$

The values for $j_c$ were set based on the results reported by Medlyn et al. (2001): coniferous species had a value of $(1 - a)$ between 0 and 0.2, whereas broadleaves had values up to 0.4. Therefore, for conifers, a value of 0.1 was selected. For broadleaves, as there seemed to be some acclimation for trees growing in elevated $CO_2$, a more conservative (than 0.4) value of 0.25 was chosen. The value for mixed forests corresponds to the arithmetic mean of the two.

*The text has been corrected and clarified as described above (p.12 l.7 and following).*

**P13. So PPo includes the carbon costs? What are these values based on?**

This variable is described in Speich et al. (2018b), and combines the plant-specific characteristics of Eq. 13 as follows:

$$PP_o = \frac{\gamma_{r,20} D_r}{L_r w_{ph}},$$

where $\gamma_{r,20}$ is the root respiration rate at 20 °C. The actual root respiration rate is dependent on annually averaged temperature via a $Q_{10}$ function, as described in Speich et al. (2018b). These details are indeed important, and will be included in the new version.

*The description of the rooting depth module has been completed (p.13 l.21 and following).*

**P16.L1-5. If a large amount is diverted by pipelines, can you compare modelled and observed discharge? Which sub-catchments are affected by this?**

The streamflow data used in this study was obtained from the company operating the power plants and includes the amount of water diverted through the different pipelines. From this, time series of natural streamflow were reconstructed, which were used as observations.

*The text has been completed as described above (p.17 l.23 and following).*

**P16.L27. "As the sampling plots... a larger area." This sentence is a bit unclear to me, what do you mean?**

Here, we explain why it does not make sense to compare simulations and observations at the scale of single inventory sampling plots (which is sometimes still being done). New formulation to clarify: „As the sampling plots of the NFI are distributed on a regular grid, each plot is randomly selected from all forest plots in that region, and may not be considered representative for a larger area. *It is therefore not sensible to compare simulated and observed biomass at the scale of single inventory plots. Instead*, the 245 NFI plots in the study area were aggregated to seven classes based on aspect and elevation,

with four elevation bands for North-facing plots and three for South-facing plots. This way, each class has a sample size of at least 30 plots, which ensures that the averages are representative."

*The text has been clarified as described above (p.17 l.28 and following).*

**P17.L19. Please correct reference**

Thank you for catching this.

**P22.L31 The plot only shows Acer spp., so how can I see this?**

This is indeed not visible on Fig. 6. However, it seems that differentiating between the three Acer species will add little value to the figure (especially as the Acer spp. band is quite thin in all plots), while making the plots harder to read. We will rephrase the text to make it clear that this is not visible on the plots.

*The text has been clarified as discussed above (p.26 l.1).*

**P24.L21. This sounds a bit counter-intuitive, wouldn't you expect a higher biomass when there is no LAI-reduction? What is the reason for this?**

An important point to keep in mind (which should be made clearer in the manuscript) is that TreeMig/FORHYCS does not explicitly simulate carbon cycling/allocation (see also response to the next point). Hence, there is no direct effect of LAI reduction on biomass. Such effects are implicitly simulated through the environmental stress functions such as Eq. 6.
Two main effects happen in the model as a result of LAI reduction: the drought index (Eq. 5) is lower than it would be without LAI reduction; and the light distribution is modified, i.e. lower height classes get more light than they would get without LAI reduction. These two effects both promote tree growth, which explains why the model simulates higher biomass. This higher growth also eventually leads to greater mortality, after the number and size of trees have grown fast for some years. This explains the more dynamic pattern when LAI reduction is activated (Fig. S9).
These effects indeed need to be discussed in the text, and will be included in Section 4.2 (Effect of coupling on forest simulations) in a revised version.

*The text now points out that there is no direct link between the assimilation partitioning described in Eqs. 8-11 and simulated biomass (p.11 l.17). The effects discussed above are now part of the Discussion in Section 4.2 (p.37 l.10 and following).*

**P29.L10-11. Is this not counter-intuitive too? You would expect (also because of equations 7 and 8, where additional carbon is allocated for roots under stress) that the roots will go deeper in case of drier scenarios, and that LAI would go down, correct?**

As noted in the response to the previous comment, there is no simulation of carbon uptake and allocation in the model. Eqs. 7 and 8 have no (direct) influence on the development of roots – they only influence the LAI reduction (which is switched off on the simulation described here). These equations were included because they represent a

sound and plausible way to parameterize reduction of leaf area due to environmental stress. This comes at the price of inconsistent formulations between Eqs. 7-11 (where carbon allocation fraction to roots is calculated as an auxiliary variable to determine the degree of leaf area reduction) and Eq. 13 (which determines rooting depth). As the allocation fractions of Eqs. 7-11 are not used anywhere else, we argue that this inconsistency can be tolerated. However, in a new version of this manuscript, we will need to make it clearer that Eqs. 7-11 only affect leaf area.

The rooting depth scheme used in FORHYCS assumes that plants dimension their rooting systems to optimize for net carbon gain. In this scheme, rooting depth does not necessarily increase with a drier climate. In some cases (like the low-elevation regions under drying scenarios, as discussed here), it may not be worth it for the plants to invest more carbon into roots. This is a point that should be included in the discussion (where a link can be made to Speich et al. (2018b), where the behavior of this rooting depth scheme was examined under various environmental conditions).

*The part of the Discussion relating to coupled modeling under climate change (Section 4.3) has been expanded to include a discussion of rooting depth (p.38 l.27 and following).*

**P31.L4-5. You described earlier that Eq. 12 was set to 1, correct?**

Only in the NCS runs – in all other runs, the effect of CO2 concentration on stomatal resistance is activated (see our response to the corresponding point above).

**P32.L4-14. How different are the landuses eventually at the end of the runs? Are the differences mainly due to different forest covers under the different scenarios?**

There is no other mechanism for land cover change in the model than forest growth or retreat. In the runs with land-cover change enabled, the forest biomass in the cells initially belonging to the „potentially forested" land cover classes reaches up to 150 t/ha at the end of the simulation (these values vary with climate scenario and elevation band). This information is indeed important to follow the presentation and discussion of these results, and will be included in the new version.

*This information is now given in the Results section (p.33 l.24); it was mentioned in the Discussion in the previous version.*

**P33.L14. Based on the data as shown, you cannot claim that the sensitivity of streamflow to vegetation properties varies spatially. This would also mean you need to have the same vegetation in different places and observed different changes in streamflow, but I believe that is not the case.**

This sentence was indeed formulated in an ambiguous/misleading way, and this is not the point that we wanted to make here (the sensitivity of water balance to absolute values of vegetation properties was the subject of another of our studies (Speich et al. 2018a)).

Rather, the point here is that the effect of model coupling (i.e. of dynamically varying the values of vegetation properties) varies spatially. This statement (which will be corrected/clarified in the new version) is directly based on the results discussed immediately before.

*This sentence has now been clarified (p.36 l.1).*

**P37.L7-8. "the greatest effects occurred at low elevations, and in regions currently above the treeline", where do you show this? Please back this up with some evidence in the main manuscript, especially when it is a key-finding.**

This is directly linked to the statement discussed in the previous point, with the greatest changes in streamflow occurring in the Chippis subcatchment (Fig. 10 and its equivalents for the other subcatchments, to be included in the supplement) and in the currently unforested, high-elevation subcatchments if forest is allowed to grow there (Fig. 11).

**Eq3. Please define all variables and subscripts**

**Table1. There two Succ_noHmax-scenarios, please correct.**

**Table 2. Please describe the abbreviations in the caption or replace them with a description.**

**Fig1a. Please define SFC also in the figure.**

OK to all

**Fig5. There are a couple of things that seem a bit odd to me in this plot. In Fig 5a, Prevah seems to be much closer to the observations then the other two model set-ups, but in Table 2 the KGE-values are lower. Is that correct? In addition, Forhycs00 and Forhycs11 are on top of each other in Figure 5a, whereas Figure 5b suggests a difference of up to 0.3 m3/s.**

What may create some confusion in this figure is that the lines in Fig. 5a are presented as 30-day rolling averages, while the line in Fig. 5b is not. For example, at the time of the greatest difference (Fig. 5b) in early 2005, the lines on Fig. 5a depart slightly but visibly from each other. If the lines on Fig. 5a were not presented as rolling averages, this difference would be more pronounced, but the plot would be more difficult to read.

*The caption of Fig.5 now mentions that Fig. 5b does not show rolling means.*

**Code availability: I would suggest to share your code on github or gitlab, instead of the supplement. Please also add links to the actual datasets used in the study.**

It is feasible to create a public repository for the model code. However, due to the different levels and structure of documentation in the code of the original models, it may be rather difficult to study the model code.
Due to restrictions from the provider of the meteorological data, it is unfortunately impossible to give access to the data used to drive the model.

*As requested by GMD's editors, the model code is now hosted in a permanent repository and publicly accessible.*

**Appendix A: Why is there an appendix in the main manuscript and also a supplement? Should Table A1 not just be part of the Supplement then?**

This can be done, it is certainly a good idea to keep the main document as lean as possible.

*This table is now part of the supplement*

**New Figures**

[Figure]

*Equivalent to Fig. 10 for the Moulin subcatchment – to be added to the Supplement.*

[Figure]

*Equivalent to Fig. 10 for the Vissoie subcatchment – to be added to the Supplement.*

[Figure]

*Equivalent to Fig. 10 for the Mottec subcatchment – to be added to the Supplement.*

[Figure]

*Equivalent to Fig. 10 for the Moiry subcatchment – to be added to the Supplement.*

[Figure]

*Modification of simulated mean annual streamflow if the effect of CO₂ concentration on stomatal resistance is enabled. To be added to the supplement.*

*This is an update to the response submitted to the comments of Reviewer #1 in the interactive discussion. This version also describes how the changes were included in the revised manuscript. All changes to the original response document are marked in italic. Page and line numbers of the new manuscript refer to the marked version.*

We would like to thank the reviewer for their thorough response to our manuscript. Their comments will be very helpful to improve our manuscript. We are glad to take the opportunity of this discussion format to address the points they raise.

**My main concern about the research is, this study area is not an ideal watershed for doing the experiments. Since we are talking about coupling a forest model with a hydrological model, we better have a catchment that is covered with forest. The five subcatchments of Navizence, have 30% forested area in Chippis (but no streamflow data) and ~15% forested area in Vissoie. The other three catchments have negligible fraction of forested area. That might be part of the reason that we don't see much hydrograph difference between uncoupled and coupled runs even in the most forested catchment Vissoie. In fact the uncoupled-coupled streamflow differences are smaller in the other less forested catchments**

The relative fractions of currently forested, potentially forested and never-forested area are shown on Fig. 2. The fractions of forested area are higher than the numbers stated in the review: more than half of the Chippis subcatchment is currently forested (> 25 km² out of ~50 km²), and for Vissoie, this number is about 25% (~15 out of ~60 km²).

The abandonment of pastures and replacement with forests is an ongoing process in this region, and is expected to continue given current socio-economic trends (see Price et al. 2016). Therefore, the potentially forested area is also relevant, and the behavior of the model for this situation is also tested in this study. For the five subcatchments of this study, the fractions of currently and potentially forested area are ~ 90% (Chippis), 66% (Torrent du Moulin), 70% (Vissoie), 25% (Mottec), 33% (Moiry/Lona).

Considering also the other reasons for selecting this study region, listed in Section 2.2 (p.14f) (strong ecological and hydro-climatic gradient due to elevation differences), we argue that this is a suitable study region for testing the model. Another study is planned in which we apply FORHYCS to several catchments representing different hydro-climatic regions of Switzerland.

**It would be great if authors could acquire downscaled meteorological data from latest climate modeling scenarios and test the FORHYC with those time series data. It's not that the delta method (e.g. ±degrees and/or precipitation) isn't scientifically sound, but the climate modeling data provides more variation and insight to the change of climate.**

In the framework of an ongoing study, we have acquired the CH2018 scenarios (NCCS 2018), based on the EURO-CORDEX simulations. Given the scope of this study, it does not seem necessary to run FORHYCS with all 39 model chains. Instead, we suggest providing results for the three chains selected by Brunner et al. (2019) as representative for dry, medium and wet conditions (in addition to the delta runs).

*The study now includes simulations with meteorological forcing from three GCM-RCM chains. These model chains are presented in Section 2.2.4 (p.21 l.23 and following). The Supplement shows how precipitation and temperature develop in these three model chains (Fig. S2-S5). The output from the FORHYCS simulations with meteorogical forcing has been added to Figs. 9 and 10.*

**P1L12-13: give names of the two new metrics.**

The names for these metrics (dH95 and 1-ABC) are introduced in Section 2.2.3.

**P2L22: what aspects of mountainous regions are sensitive to what type of global change?**

We will be more specific on this in a revised version and mention the following aspects:

- Strong influence of snow storage on hydrology, which may change with warming.
- Warmer and drier conditions may alter species composition in temperature-limited ecosystems.
- Upwards shift of treeline.

*This has been completed (p.2 l.24).*

**P8L7: explain Al, is it the stand leaf area?**

Yes – we will include this in the text.

*This has been clarified (p.8 l.8).*

**P9L3: define EDI, is it Evaporative Demand Index?**

This is a remainder of an old version of the text, thank you for catching this. This refers to the drought index used in TreeMig, and described in the rest of the paragraph. The symbol "EDI" will be removed.

*This has been corrected (p.9 l.25).*

**P10L5: again, define fDS.**

Vitality reduction function due to drought stress. Based on the comments of Reviewer #1, this passage will need to be clarified. We will also consider this remark.

*The description of the vitality reduction functions has been clarified (p.10 l.11 and following).*

**P14L7: forest minimum stomatal resistance at 180 s m-1 seems to be at the lower end of what has been reported. The stomatal/canopy resistance could be one of the most influential parameters when it comes to estimating transpiration (not sure about this PREVAH model). Why is a single number resistance superior to the previous "minimum canopy resistance for each land cover class"? In any case, the**

**number needs to be justified according to the region and species being applied in this particular study.**

This concerns only uncoupled FORHYCS, i.e. a simultaneous run of the hydrological and forest parts of the model without any transfer of information between them. For coupled FORHYCS, minimum stomatal resistance is parameterized according to species and tree height, as described in Section 1.3.2 of the Supplement (which we forgot to reference from the main text – this will be fixed in a new version).

Uncoupled FORHYCS is not meant to represent an improvement from stand-alone PREVAH. Rather, its purpose here is to provide a comparable baseline against which to assess the effect of the coupling. This was not done directly against stand-alone PREVAH for the reason stated in Section 2.1.7 (PREVAH uses a different dataset for soil water storage capacity which is incompatible with the new water balance module).

In another study (Speich et al. 2018a), we found that RSMIN was a rather influential vegetation property for simulated annual water balance (though not as influential as soil water storage capacity or LAI). This was one of the motivations for coupling the models. For uncoupled FORHYCS, the value of 180 s/m was selected as a standard value from the literature (see the cited reference of Guan and Wilson 2009, who base their choice on the review by Körner (1994)). For comparison, in another study in a region nearby (Peters et al. 2019), stomatal resistances of 128 s/m and 285 s/m were found for *Larix decidua* and *Picea abies*, respectively.

**P22L9-23: I've got confused by the streamflow simulations. How was the model calibrated to generate these KGE scores? I assume the PREVAH/FORHYCS were "tuned" to their best performance before the series of experiment. It seems three modeling runs generated model efficiencies that vary from catchment to catchment. Would some other combinations of parameterization make the results look differently?**

As Reviewer #1 had a similar comment, here is our answer to their question regarding calibration:

Indeed, there are various parameters related to non-vegetation aspects. Some of these parameters are constant for the whole study area; others are spatially variable and have a different value for each grid cell. The spatially variable parameter values were determined in a previous study (Zappa and Bernhard 2012) based on the regionalization method of Viviroli et al. (2009). The spatially constant parameters were also taken from previous studies. As a different dataset for soil water holding capacity was used in this study (see Section 2.2.2), some parameters were manually adjusted to improve the optical fit of the streamflow lines. Due to the proof-of-concept nature of this study, a full calibration was not undertaken.
In a revised version, we will provide the reference for the spatially variable parameter values, as well as the values used for the spatially constant parameters, in the Supplement.

**P23-Figure 5: I'd suggest giving PREVAH a solid light grey line to make it easier to read the difference between FORHYCS00 and FORHYCS11 (which matters more).**

OK

**P26L8-20: labels from different simulations need to be unified. For example, Succ_TM_BEK is BEK in the Figure 8?**

Based on comments from Reviewer #1, we will reconsider the naming of the different runs. We will also take this comment into account.

*The names of runs have been revised (Table 1) and the use of run names is now consistent throughout the manuscript. Also, a new paragraph explains the nomenclature for the different model runs (p.19 l.19).*

**P33L3-15: looks like a large part of the reason that uncoupled and coupled modeling runs were making rather subtle differences is, the catchment areas are not forested enough for the forest model to pass signals back to the hydrology module. If we look at the hydrologic modeling performance for catchments, Vissoie has more vegetated areas (other than Chippis, which has no observations), thus has the lowest KGE score. The other three catchments, none with meaningful forest fraction, perform much better without interplay with the forests.**

It is plausible that the degree of forest cover had some influence for the KGE scores. However, I would also see it in the view of the different soil parameterizations used in PREVAH and uncoupled FORHYCS. Those differ in currently and potentially forested areas, so that lower performance is to be expected in those areas.
It is also worth considering that streamflow from the Mottec and Moiry subcatchments stems to a large extent from the glaciers, the water balance of which the model is able to simulate relatively well.

[revised manuscript text omitted]

---

## Referee Report (RR1)

**Review of "FORHYCS v1.0:** A spatially distributed model combining hydrology and forest dynamics" by Speich et al.**

I am happy that my previous review proved useful for the authors. I think the manuscript improved a lot compared to the previous version and the authors addressed my major concerns adequately. However, when reading over it again in detail, I still encountered quite some minor points. I think these should be fairly easy to address and hope these remarks proof useful again for the authors.

**Minor comments**

P1.L14. It is not clear here what a "delta change" approach is.

P10.L11. "This reduction function"  $\rightarrow$  the environmental reduction function or the shading reduction? P10.L24. Please define DDEGS

P15.L20. (2) Effects -> , (2) effects

P17.L17. Acconuting  $\rightarrow$  accounting

P17.L16-18. I would suggest to elaborate a bit on how the timeseries were constructed. The reader has to guess now how reliable it actually is. I think it would also be good to add the lake and diversions in the map. I think it is important to do this, as comparing a reconstructed time series with a modelled time series does not seem very reliable to me, and I agree with the authors it can only be used to plausibilize the outcomes.

P17.L20. Remove "a"

P19.L15-L16. "In this...mode ("F")  $\rightarrow$  this is partly a repetition of L10-L11 "The names... with "S"." P19.L17. What does \_NCS stand for?

P19.L19. Suuccession –> succession

P23.L2. Why was this the best?

P23.L5. What is the "delta" set?

P23.L31. (Gupta et al., 2009, KGE,)  $\rightarrow$  remove additional comma

P24.L2. Table 2  $\rightarrow$  Table 3

P26.L8-9. Thank you for the addition, but you can't make this statement and not show the evidence.

Show either the data, or remove your statement.

P28.L1. On  $\rightarrow$  in?

P25.L16. You mean the average LAI-values, correct?

P30.L2-3, Figure 8. S\_F\_Full and S\_F\_noSmort are not visible in the figure. You could change the linetype and/or (size of) the symbol to make it visible.

P31.L8. n particular  $\rightarrow$  In particular

P31. Please use the same scenario-names as in Table 1 and Fig. 9., i.e. C\_T\_BEK and C\_T\_RA15 instead of TM\_BEK and TM\_RA2015

P31.L24-25. Inter-annual ...TM\_BEK (Fig. S17 c)).  $\rightarrow$  Can you clarify how I can see this? Are you still discussing the dry-scenario?

P33.L16. Differences of up -> differences up to

P36.L4."require" -> Don't you mean "lead to"?

P36.L6-7. "Any case ... uncoupled models" -> You discuss canopy structure, so Figure 7, but how can you relate that to water availability and temperature? Does this relate to the soil parameterizations of BEK and RA2015?

P37.L15. Fig. S1 only shows soil moisture capacity and difference in drought stress, there are no LAI-values.

P36.L36. in this study in the runs  $\rightarrow$  in this study are in the runs

Fig.7. The xlabel shows a square, but I think this is a delta.

Fig.8. Please add units and labels to the figure.

Fig.9. The legend has both dry, medium, wet and dP-10%, dP-0%, dP+10%, maybe clarify that dry, medium and wet relate to GCM-RCM.

I also have to say that I still find the article sometimes hard to read, because of all the case names that are not directly clear to the reader. This is probably a matter of personal taste, but I think it may help if the authors occasionally write out the scenario in paragraphs where it is discussed.

---

## Author Response (AR2)

We would like to thank the reviewer for once again taking the time to read our manuscript and give us the opportunity to improve our article. Here is our point-by-point response:

**P1.L14. It is not clear here what a "delta change" approach is.** This has now been replaced with „idealized temperature and precipitation change"

**P10.L11. "This reduction function" → the environmental reduction function or the shading reduction?** Clarified as „The environmental reduction function"

**P10.L24. Please define DDEGS** Added (degree-day sum; see Section 1.2 of the Supplement)

**P15.L20. (2) Effects –> , (2) effects** OK

**P17.L17. Acconuting → accounting** OK

**P17.L16-18. I would suggest to elaborate a bit on how the timeseries were constructed. The reader has to guess now how reliable it actually is. I think it would also be good to add the lake and diversions in the map. I think it is important to do this, as comparing a reconstructed time series with a modelled time series does not seem very reliable to me, and I agree with the authors it can only be used to plausibilize the outcomes.** The new Section 1.9 of the Supplement explains the nature of the streamflow data and provides a reference for further information (including the location of the lake and origin of diverted water).

**P17.L20. Remove "a"** OK

**P19.L15-L16. "In this…mode ("F") → this is partly a repetition of L10-L11 "The names… with "S"."** I do not understand this point. L15-16 refer to the second series of experiments (the „C_" simulations), while L10-11 refer to the first series of experiments.

**P19.L17. What does _NCS stand for?** Clarified (NCS; standing for "No $CO_2$ effect on Stomatal resistance")

**P19.L19. Suuccession –> succession** OK

**P23.L2. Why was this the best?** Clarified: „as it produced the most plausible long-term biomass dynamics (see Sect. 3.2.1) and shows a good fit to observed canopy structure (see Sect. 3.2.2)"

**P23.L5. What is the "delta" set?** Modified to „idealized climate change runs"

**P23.L31. (Gupta et al., 2009, KGE,) → remove additional comma** OK

**P24.L2. Table 2 → Table 3** OK

**P26.L8-9. Thank you for the addition, but you can't make this statement and not show the evidence. Show either the data, or remove your statement.** The statement on the elevational distribution of the two maple species has now been removed.

**P28.L1. On → in?** OK

**P25.L16. You mean the average LAI-values, correct?** Yes, clarified as „the average forest LAI"

**P30.L2-3, Figure 8. S_F_Full and S_F_noSmort are not visible in the figure. You could change the linetype and/or (size of) the symbol to make it visible.** The lines for these two cases are now set bolder and are indeed more visible.

**P31.L8. n particular → In particular** OK

**P31. Please use the same scenario-names as in Table 1 and Fig. 9., i.e. C_T_BEK and C_T_RA15 instead of TM_BEK and TM_RA2015** OK

**P31.L24-25. Inter-annual …TM_BEK (Fig. S17 c)).→ Can you clarify how I can see this? Are you still discussing the dry-scenario?** This statement refers to the idealized climate change runs, and adding the reference to the GCM-RCM chains has isolated it from its contexxt. This sentence has now been moved to make it clear which runs are discussed.

**P33.L16. Differences of up –> differences up to** OK

**P36.L4."require" –> Don't you mean "lead to"?** I think that both formulations are valid, depending on how one sees the modeling process. To avoid any ambiguity, we have replaced this sentence with „The species-specific drought tolerance parameters used with one index cannot be used with the other."

**P36.L6-7. "Any case … uncoupled models" –> You discuss canopy structure, so Figure 7, but how can you relate that to water availability and temperature? Does this relate to the soil parameterizations of BEK and RA2015?** Water availability are the two processes that modify species-specific maximum tree height. This sentence has been clarified by adding „, as seen by the better fit to observed canopy structure for the runs which include this effect (Fig. 7)"

**P37.L15. Fig. S1 only shows soil moisture capacity and difference in drought stress, there are no LAI-values.** Indeed, Fig. S1 only shows soil moisture storage capacity, and the link to LAI is made in the following sentences. Syntax has been modified to avoid any confusion.

**P36.L36. in this study in the runs → in this study are in the runs** I do not understand this point. There is no L36 on that page and I cannot find this word sequence in the document.

**Fig.7. The xlabel shows a square, but I think this is a delta.** Strange, this is displayed correctly on my work computer (Mac) and other devices. I have now saved this figure as a PNG, so it should work fine anyway.

**Fig.8. Please add units and labels to the figure.** OK

**Fig.9. The legend has both dry, medium, wet and dP-10%, dP-0%, dP+10%, maybe clarify that dry, medium and wet relate to GCM-RCM.** OK; Figs. 9 and S17 have been clarified.

**I also have to say that I still find the article sometimes hard to read, because of all the case names that are not directly clear to the reader. This is probably a matter of**

**personal taste, but I think it may help if the authors occasionally write out the scenario in paragraphs where it is discussed.**

We agree that it can be difficult to relate the run names to the properties of the different cases. Therefore, we provided a short description of all runs in Sect. 3.2.1, as well as an occasional reminder for individual runs later in the text.

[revised manuscript text omitted]